

# 1 In situ observations of supercooled liquid water clouds over Dome

# 2 C, Antarctica by balloon-borne sondes

**Philippe Ricaud[1], Pierre Durand[2], Paolo Grigioni[3], Massimo Del Guasta[4], Giuseppe**
**Camporeale[5], Axel Roy[1], Jean-Luc Attié[2], and John Bognar[6]**
[1]CNRM, Université de Toulouse, Météo-France, CNRS, 42, Avenue G. Coriolis
31057, Toulouse Cedex, France
[2]Laboratoire d'Aérologie, Université de Toulouse, CNRS, UPS, 14 Avenue Edouard Belin,
31400, Toulouse, France
[3]ENEA, Laboratory for Observations and Measurements for Environment and Climate, Via
Anguillarese, 301 00123, Rome, Italy
[4]INO-CNR, Via Nello Carrara, 1 – 50019 Sesto Fiorentino, Italy
[5]IREA – CNR, Via G. Amendola n. 122 D/O, 70126 Bari, Italy
[6]Anasphere, Inc., 5400 Frontage Road, 59741Manhattan, MT, USA
Correspondence: philippe.ricaud@meteo.fr





**Abstract**
Clouds in Antarctica are key elements that affect radiative forcing and thus Antarctic climate
evolution. Although the vast majority of clouds are composed of ice crystals, a non-negligible
fraction is constituted of supercooled liquid water (SLW, water held in liquid form below 0°C).
Numerical weather prediction models have a great difficulty to forecast SLW clouds over
Antarctica favouring ice at the expense of liquid water, and therefore incorrectly estimating the
cloud radiative forcing. Remote sensing observations of SLW clouds have been carried out for
several years at Concordia station (75°S, 123°E, 3233 m above mean sea level), combining
active LIDAR measurements (SLW cloud detection) and passive HAMSTRAD microwave
measurements (liquid water path, LWP). The present project aimed at in situ observations of
SLW clouds using sondes developed by the company Anasphere, specifically designed for
SLW content (SLWC) measurements. These SLWC sondes were coupled to standard
meteorological pressure-temperature-humidity sondes from the Vaisala Company and released
under meteorological balloons. During the 2021-2022 summer campaign, 15 launches were
made, of which 7 were scientifically exploitable. Above a height of 400 m above ground level,
we found that the SLWC sondes detected SLW clouds in a vertical range consistent with
LIDAR observations. In nominal operation, the LWP values obtained either by HAMSTRAD
or vertically-integrated from the SLWC sonde profiles were consistent in spite of their low
values (< 10 g m$^{-2}$). On some occasions far from nominal operation (surface fog, low vertical
ascent of the balloon), the LWPs from the SLWC sonde were overestimated by a factor of 5-
10 compared to the HAMSTRAD values. In general, the SLW clouds were observed in a layer
close to saturation (U > 80%) or saturated (U ~100-105%) just below or at the lowermost part
of the entrainment zone or capping inversion zone which exists at the top of the Planetary
Boundary Layer and is characterized by an inflection point in the potential temperature vertical



profiles. Our results are consistent with the theoretical view that SLW clouds form and pertain
at the top of the Planetary Boundary Layer.


## 1. Introduction

Clouds in Antarctica are key parameters that affect the Earth radiative balance thus the climate evolution over Antarctica but also over the Earth through complex teleconnections (Lubin et al., 1998). The nature of the clouds (ice or liquid or mixed phase, a mixture of liquid and solid water) and their vertical distributions together with their interactions with aerosols add complexity to this topic. Numerical simulations at local or global scales, focused on short time scales or climate evolution show large differences between clouds located above the Southern Ocean, the Western Antarctica – and particularly the Antarctic Peninsula –, the Eastern Antarctic Plateau and in fine Antarctic coastal areas. In general, ice clouds are relatively well estimated by the models while supercooled liquid water (SLW) clouds tend to be underestimated because the water partition function favours solid instead of liquid phase for temperature less than 0°C. This flaw is rather observed in global-scale models but could be reduced in models including a detailed microphysics scheme (e.g. Engdahl et al., 2020). Therefore, the impact of the clouds on the net surface radiation, the so-called cloud radiative forcing, that strongly depends on the nature of the cloud, is usually underestimated by 5-30 W m$^{-2}$ in models that favour ice instead of SLW clouds (King et al., 2006, 2015; Bromwich et al., 2013; Lawson and Gettelman, 2014; Listowski and Land-Cope, 2017; Young et al., 2019). From observations and climate models, it appears that, in Antarctica, the liquid water path (LWP), which is the vertically-integrated SLW content (SLWC), is on average less than 10 g m$^{-2}$, with slightly larger values in summer than in winter by 2-5 g m$^{-2}$ (Lenaerts et al., 2017), whereas, in the Arctic, values greater than 50 g m$^{-2}$ were reported (Lemus et al., 1997; Zhang et al., 2019) and, at middle/tropical latitudes, values ranging 100-150 g m$^{-2}$ were measured and modelled (Lemus et al., 1997).

In parallel, cloud observations over Antarctica are difficult because of the very small number of ground stations which are located preferably near the coast with only three of them



opened all year-long deep inside the continent. It is the reason why space-borne measurements
are paramount to classify clouds over the entire continent as a function of height, nature, and
time. It is clearly accepted now that SLW clouds are much more abundant near the coast than
in the inner continent (Bromwich et al., 2012; Listowski et al., 2019) with larger ice crystals
and water droplets (Lachlan-Cope, 2010; Lachlan-Cope et al., 2016; Grosvenor et al., 2012;
O'Shea et al., 2017; Grazioli et al., 2017) and that the cloud radiative forcing is maximum over
the Antarctic Peninsula with values reaching 40 W m$^{-2}$ (Ricaud et al., 2024). In addition to this
continent-scale information provided by satellites, it is crucial to obtain information at the local
scale from remote and/or in situ observations. Remote observations of SLW/mixed phase cloud
are usually performed by means of backscattered LIDARs and ceilometers while in situ
observations have been performed over the Southern Ocean (Chubb et al., 2013), Western
Antarctica (Grosvenor et al., 2012; Laclan-Cope et al., 2016) and coastal areas (O'Shea et al.,
2017) using instruments on board aircraft.
At Concordia station, several studies from remote-sensed observations already took place
to evaluate: 1) the presence of the SLW/mixed phase clouds over the station mainly based on a
backscattered LIDAR (Cossich et al., 2021), 2) the amount of the LWP within SLW clouds, 3)
the impact of SLW clouds on the net surface radiation (Ricaud et al., 2020), 4) the differences
between observations and model simulations of SLW clouds, 5) the relationship between in-
cloud temperature and LWP, and 6) the relationship between LWP and cloud radiative forcing.
In general, SLW clouds are preferably observed in summer with very small LWPs (< 10 g m$^{-2}$
$^{2}$), in-cloud temperatures ranging from -20°C to -38°C and a cloud radiative forcing up to a
maximum value of 40 W m$^{-2}$ (Ricaud et al., 2024).
We have thus proposed a new project to observe SLW clouds in situ at Concordia, based
on the use of a sonde developed by the Anasphere Company and especially designed for the
detection of this type of cloud. During the summer campaign 2021-2022, the SLWC sonde was
connected to a standard Vaisala pressure-temperature-humidity (PTU) sonde and embarked
under an ascending balloon while, during the summer campaign 2022-2023, the two coupled
sondes were installed aboard a vertical take-off and landing (VTOL) drone. Numerous SLW
clouds were present during the 2021-2022 campaign while, in 2022-23, they were very scarce
over the station with a net consequence of measuring only vertical profiles of temperature and
relative humidity (Ricaud et al., 2023).
The aim of the present study was to perform for the first time in-situ observations of SLW
clouds above the Concordia station during the summer campaign 2021-2022. For the validation
and interpretation of the data, we relied on the observations performed by 1) the backscatter
LIDAR installed at the station for more than ten years to characterize the nature of the cloud
(ice/liquid/mixed phase) and its height and 2) the LWPs measured by the HAMSTRAD
radiometer set up at the station in 2009.
The article is structured as follows. The instruments are presented in Sect. 2. The
methodology is explained in Sect. 3. The results of the campaign are presented in Sect. 4 before
being synthetized and discussed in Sect. 5. A conclusion finalizes the findings in Sect. 6. Note
that all the observations performed during the summer campaign are presented in a companion
document as supplementary materials.

**2. Instruments**
In addition to the Vaisala PTU and Anasphere SLW sondes attached to the meteorological
balloons, we used observations from two other instruments installed at the Concordia station
for several years, namely the backscatter LIDAR to classify the cloud as an SLW cloud, and
the HAMSTRAD microwave radiometer to obtain the LWP.




*2.1. PTU sondes*
The PTU sondes used during the 2021-2022 summer campaign were standard Vaisala RS-
41 SGP sondes (an upgraded version of the Vaisala's RS92 radiosondes), which are now used
daily at Concordia to obtain operational temperature and humidity vertical profiles at 12:00
UTC. The sondes were attached to the balloon with a string either unwound before launching
(and with a length $L$ = 20 or 40 m) or wound on an unwinder. We systematically used a
parachute to obtain vertical profiles in both the ascending and descending phases.
*2.2. SLWC sondes*
The Anasphere's vibrating-wire sonde records a vibrating wire's frequency as ice
accumulates along its length (Serke et al., 2014). These frequency measurements, combined
with collocated meteorological measurements, can be used to determine the SLWC of the
surrounding air. The SLWC sonde actually measures the frequency of the vibrating wire. Since
this frequency $f$ varies according to the change in mass of the wire, its derivative with respect
to (wrt) time $df/dt$ can be used to calculate the water collected by the wire, either in the form
of ice or absorbed liquid, depending on whether the wire in question is gel-coated or nickel-
plated, respectively. From Dexheimer et al. (2019), SLWC (g m⁻³) is estimated to be:
$$SLWC = -(2b_0 f_0^2 / \varepsilon D \omega f^3) \times (df/dt) \tag{1}$$
where $\varepsilon$ is the droplet collection efficiency (~0.9), $D$ is the wire diameter including the
hydrophilic gel (0.030 inch or 0.762 mm), $b_0$ is the vibrating-wire mass per unit length
including the hydrophilic gel (2.24 g m⁻¹), $\omega$ is the velocity of air relative to the wire (~5 m s⁻
¹) and $f_0$ is the un-iced wire frequency in Hertz ranging from 21.50 to 22.50 Hz during the
campaign. $f$ typically ranges from 20.0 to 22.85 Hz during the campaign. The output signal of
the sonde is connected to the Vaisala radiosonde which transmits the data to the ground station
via telemetry. The observations of the two sondes are thus synchronized. The integration time
is 5 s, thus providing an observation every ~25 m along the vertical. We have applied a 4-point





running average to all our observations. This means that our vertical profiles, even sampled
every ~25 m, have a vertical definition of about 100 m. Since it takes about 60-80 s from the
launch for the SLWC sonde to stabilize, the minimum height for meaningful observations is
~300-400 m above ground level (agl), below which we are unable to detect any SLW cloud.
Note that, in the following, all heights are given in agl.
*2.3. LIDAR*
The tropospheric depolarization backscatter LIDAR (532 nm) has been operating at Dome
C since 2008 (see http://lidarmax.altervista.org/englidar/_Antarctic%20LIDAR.php). The
LIDAR provides 5-min tropospheric profiles of aerosols and clouds continuously, from 20 to
7000 m, with a resolution of 7.5 m. LIDAR depolarization (Mishchenko et al., 2000) is a robust
indicator of non-spherical shape for randomly oriented cloud particles. A depolarization ratio
below 10% is characteristic of SLW clouds, while higher values are produced by ice particles.
The potential ambiguity between SLW cloud and oriented ice plates is avoided at Dome C by
operating the LIDAR 4° off-zenith (Hogan and Illingworth, 2003).
*2.4. HAMSTRAD*
HAMSTRAD is a microwave radiometer that profiles water vapour, liquid water and
tropospheric temperature above Dome C. Measuring at both 60 GHz (oxygen molecule line
($O_2$) to derive the temperature) and 183 GHz ($H_2O$ line), this unique, state-of-the-art radiometer
was installed on site for the first time in January 2009 (Ricaud et al., 2010). Measurements from
the HAMSTRAD radiometer allow the retrieval of vertical profiles of water vapour and
temperature from the ground to 10 km altitude with vertical resolutions of 30 to 50 m in the
Planetary Boundary Layer (PBL), 100 m in the lower free troposphere and 500 m in the upper
troposphere-lower stratosphere. The LWP ($g\ m^{-2}$) can also be estimated. The time resolution is
adjustable and fixed at 60 seconds since 2018. Note that an automated internal calibration is
performed every 12 atmospheric observations and takes about 4 minutes. Consequently, the



atmospheric time sampling is 60 seconds for a sequence of 12 profiles, and a new sequence
starts 4 minutes after the end of the previous one. The temporal resolution of the instrument
allows the detection of clouds and diamond dust (Ricaud et al., 2017) together with the SLW
clouds (Ricaud et al., 2020). The 2021-2022 and the 2022-2023 summer campaigns were
dedicated to in-situ observations of SLW clouds using balloons and drone (Ricaud et al., 2023),
respectively. Comparisons with numerical weather prediction models showed consistent
amounts of LWP at Dome C when the ice-liquid water partition function favours SLW for
temperatures below 0°C (Ricaud et al., 2020).

**3. Methodology**

In order to optimize in-situ SLW cloud observations, we developed the following

procedure. 1) The remotely-sensed and real-time observations of clouds (either ice crystals
and/or SLW) from the LIDAR were checked regularly. 2) When the presence of SLW was
verified, we checked the value of LWP from HAMSTRAD. An empirical value of $LWP_0 = 1.0$
g m$^{-2}$ was estimated as the threshold above which an SLW cloud is considered as significant.
For $LWP < LWP_0$, either the amount of liquid water in the cloud was too low or the SLW cloud
was too scattered. 3) If the two-above conditions were fulfilled for more than 2 hours, we started
the connection and calibration process of the 2 sondes (PTU and SLWC) via the Vaisala
Digicora station inside the Concordia station. Then we went outside and inflated the
meteorological balloon. Finally, we launched the 2 sondes attached to the balloon using either
an unwinder or an unwound string (Figure 1). In total, the step 3) lasted about 1 hour. As we
used standard meteorological balloons (Totex TA100), we were able to probe the atmosphere
from the surface up to about 12-13 km height (ascent and descent) for a total duration of about
1 hour and 40 minutes. Since the tropopause height was ranging 7-8 km and we were only
interested in the first 2 km where the SLW clouds are located, only 2-5% of the observations





made were scientifically sound for our project. This is the main reason why we used a drone
during the next campaign 2022-2023 to detect SLW clouds in the PBL (Ricaud et al., 2023).
Note that, since there was only one Vaisala Digicora station for both our project and the
operational meteorological sounding at 12:00 UTC, we could not use the time window between
09:00 and 14:00 UTC for our studies.

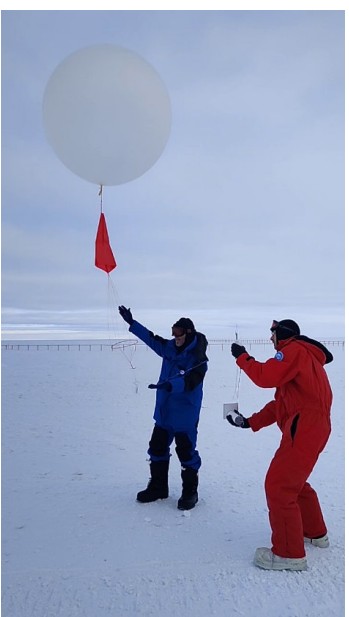


**Figure 1:** Launch of a Vaisala PTU sonde (left hand of the man in blue) and an Anasphere
SLWC sonde (right hand of the man in red) attached to the Totex TA100 meteorological
balloon, together with the red parachute and the unwinder for the first flight on 22 December

2021.


In general (see e.g., Ricaud et al., 2020), SLW clouds are usually capped by a thin

temperature inversion and a decrease from high relative humidity $U$ (>80%). As this inversion
layer separates two layers where temperature decreases with height, it contains an inflection
point in the temperature (or potential temperature) profile the height of which $H(T_{inf})$ can be

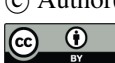

used as the top of the atmospheric boundary layer with its capping SLW cloud layer. Such a
definition based on the height of the inflection point is frequently used for the determination of
the boundary-layer thickness (Hennemuth and Lammert, 2006). Consistent with this definition,
Ricaud et al. (2020) adapted from Stull (2012) proposed to consider the potential temperature
vertical distribution separating the diurnal variation of the top of the planetary boundary layer
into 2 phases: 1) the entrainment zone at the top of the mixed layer where the SLW cloud
develops and 2) the capping inversion zone under which the SLW cloud still persists at the top
of the residual quasi-mixed layer. The vertical limits of these two layers are well defined by the
height of the inflection points $H(\theta_{inf})$. In the following, we have used information from
profiles of the potential temperature $\theta$ (K) defined as:
$$\theta = T(P_0/P)^{R/C_p} \tag{2}$$

where $T$ is the temperature (K), $P$ the pressure (hPa), $P_0$ the reference pressure (1000 hPa), $R$
the gas constant of air (J kg$^{-1}$ K$^{-1}$) and $C_p$ the heat capacity at constant pressure (J kg$^{-1}$ K$^{-1}$).
$R/C_p$ is taken at 0.286. We have characterized inflection points heights $H(\theta_{inf})$ in the potential
temperature vertical profiles when the second derivatives in $\theta$ with respect to the height $z$
$(d^2\theta/dz^2)$ are greater than an empirical threshold value typically varying from 1.5 10$^{-4}$ to 4.0
10$^{-4}$ K m$^{-2}$.

**4. Results**
*4.1. Period of study*
The balloon-borne observations of SLW clouds were carried out during the 2021-2022
summer campaign at Concordia. A total of 15 launches were performed from 21 December
2021 to 28 January 2022 (labelled from L01 to L15, respectively). With the exception of 17
January 2022 (L11), when the observations were made to check the behaviour of the SLWC
sondes in cloud-free conditions, all other launches were made when a SLW cloud was detected



for more than 2 hours with the LIDAR observations using the depolarization method described
in section 2.3.

Table 1 lists all the launches that were scientifically exploitable in ascending, descending

or both modes. In order to avoid listing a catalogue of observations, we chose to only show
details and Figures relative to the launches performed on 25 December 2021 and on 17 January
2022 (cloud-free period). Nevertheless, the SLWC vertical distributions calculated for all the
flights are shown and discussed in the forthcoming sections. The information regarding all the
flights are presented in the supplementary materials. This encompasses: 1) the LWP values
from HAMSTRAD and the height range of the SLW clouds from the LIDAR over one day, 2)
the profiles of temperature, potential temperature and relative humidity measured by the PTU
sonde during the flights, and 3) the profile of the SLWC sonde frequency $f$, the derivative of
the frequency wrt time $t$ ($df/dt$) and the calculated SLWC during the flights.

**Table 1:** List of SLW cloud flights performed during the 2021-2022 season over Concordia,
together with date, launch time (UTC) and in italic the time (UTC) when the balloon hits the
ground after the descent, SLW cloud vertical range (m) and associated LWP (g m$^{-2}$) in
ascending (ASC) or descending (DES) phase, considering only observations above 400 m agl.
Also shown are the SLW cloud vertical range (m) observed by the LIDAR in time coincidence
with the flight and the minimum-maximum LWP (g m$^{-2}$) measured by HAMSTRAD for the
same date over 24 hours. Also included are: heights (m) of the inflection point in the vertical
profile of potential temperature $H(\theta_{inf})$, information on the type of string used (unwinder or
unwound string of length $L$), and any other relevant information (vertical ascent velocity ω less
than the nominal 5 m s$^{-1}$, cloud-free period, surface fog). Heights are always given in meters
agl. The root mean square error (RMSE) $\sigma$(g m$^{-3}$) associated with the SLWC profiles in cloud-
free conditions is also estimated.


| Launch #<br>ASC/DES | Date<br>YYMMDD | Launch<br>Time<br>HH:MM:SS<br>UTC | Comments | $H(\theta_{inf})$<br>m | SLW cloud<br>vertical domain | | LWP<br>g m$^{-2}$ | |
|---|---|---|---|---|---|---|---|---|
| | | | | | Sonde<br>m | LIDAR<br>m | Sonde | Hamstrad<br>Min-Max |
| L01<br>ASC | 211222 | 02:24:30 | Unwinder | 710-750 | 400-500 | 400-600<br>750 | 7.37 | 2-10 |
| L03<br>DES | 211225 | 08:53:15<br>*10:30:00* | Unwinder | 950-1000<br>1450-1500 | 900-1000<br>1400-1500 | 600-800<br>800-1000<br>1200-1300 | 3.67 | 2-6 |
| L04<br>ASC | 211225 | 15:48:51 | Unwinder | 850-880<br>1400<br>1520 | 700-900<br>1500 | 700-900 | 9.08 | 2-6 |
| L06<br>ASC | 211229 | 13:45:00 | $L = 40$ m<br>$H > 750$ m | < 750 | 750-850 | 600-800 | 7.48 | 1.0-3.5 |
| L07<br>ASC | 211229 | 17:47:51 | $L = 40$ m<br>$\omega \sim 3.5$ m s$^{-1}$ | 700<br>850 | 400-600<br>750-900 | 600-800 | 33.17<br>23.94 | 1.0-3.5 |
| L14<br>ASC | 220124 | 13:51:05 | $L = 20$ m<br>Fog | 630<br>900-920<br>1400 | 600<br>800-1000 | 800 | 575.35 | 1-5 |
| L14<br>DES | 220124 | 13:51:05<br>*15:30:00* | $L = 20$ m<br>Fog | 810<br>1340<br>1420 | 800<br>1000 | 800 | 18.92 | 1-5 |
| L15<br>ASC | 220128 | 06:08:27 | $L = 20$ m | 650<br>910<br>1080 | 600-800<br>1000-1100 | 600-800<br>900-1000 | 10.15<br>7.31 | 2-5 |
| L11<br>ASC | 220117 | 06:35:15 | $L = 40$ m<br>Cloud Free | | | | ~0<br>$\sigma \sim 0.05$ g m$^{-3}$ | 0.4-1.0 |
| L11<br>DES | 220117 | 06:35:15<br>*08:20:00* | $L = 40$ m<br>Cloud Free | | | | ~0<br>$\sigma \sim 0.05$ g m$^{-3}$ | 0.4-1.0 |


*4.2. Launches on 25 December 2021*
As on 25 December 2021, SLW clouds observed by LIDAR were almost continuously
present over Concordia from 00:00 to 19:00 UTC (Figure 2), 2 launches were performed at
08:53:15 (L03) and 15:48:51 UTC (L04), from which we will consider both the descending and
ascending phases. For 2 hours before the first launch, SLW clouds were observed between 500
and 700 m, and during the flight, the SLW clouds were located between 600 and 800 m with
some traces of SLW clouds between 1200 and 1300 m, while approximately 2 hours after the
flight (when the sondes hit the ground in the descending phase) the SLW clouds were located
between 800 and 1000 m. Regarding the second flight, for the 2 hours before the flight, SLW
clouds were observed between 700 and 1000 m and, during the flight, around 700-800 m. The
first launch was associated with HAMSTRAD-observed LWP of 1.5-6.0 g m$^{-2}$ whereas, for the
second flight, it was in the range 1.5-3.0 g m$^{-2}$. Note that when the sondes reached the ground





at the end of the first launch, the balloon had travelled a distance of about 70 km from the
Concordia station after a flight time of 1 h 40 min (Figure 3).

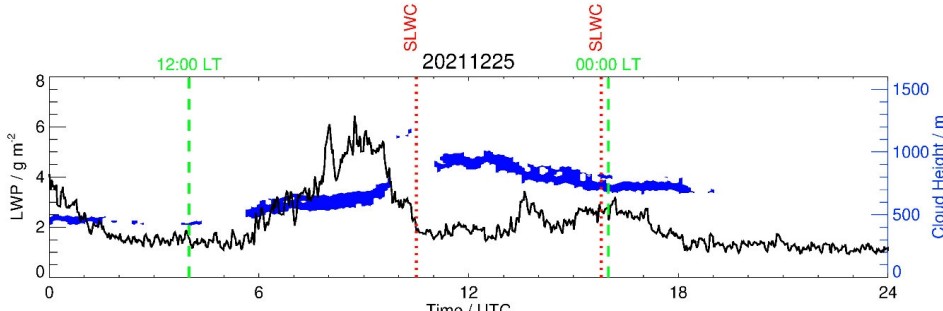


**Figure 2:** Diurnal variation of the Liquid Water Path (LWP) measured by HAMSTRAD (g m$^{-2}$, black solid line) on 25 December 2021 (UTC Time). Superimposed is the SLW cloud thickness (blue area) derived from the LIDAR observations (blue y-axis on the right). Two vertical green dashed lines indicate 12:00 and 00:00 LT. The two red vertical dotted lines indicate the ground landing of the first SLWC sonde (L03 flight) at about 10:30 UTC and the launch of the second SLWC sonde (L04 flight) at about 15:50 UTC, respectively.




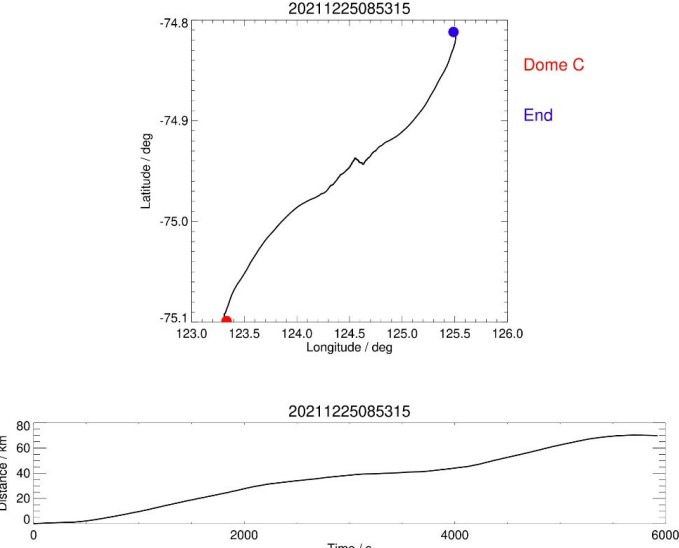


**Figure 3:** (Top) Path followed by the meteorological balloon launched on 25 December 2021 at 08:53:15 UTC (L03) (red circle) up to the end of the flight (blue circle). (Bottom) Distance travelled (km) as a function of time since launch.


In general, all the flights reached a top height above 10 km (Figure 4 and Figures S7-14), namely well above the tropopause height (about 7-8 km). This is consistent with previous observations made with meteorological operational Vaisala PTU sondes (Tomasi et al., 2015). The profiles of temperature and relative humidity measured during the whole flight (L03) starting at 08:53:15 UTC are shown in Figure 4 together with the calculated potential temperature and observed relative humidity within the layer [400-1600 m]. Above 2 km, a good consistency between ascending and descending phases is found in temperature profiles within ±1 K. The relative humidity profiles are within ±5% of each other, except between 7 and 8.5 km where they differ by around 10%. Below 2 km, the profiles reflect the impact of the PBL. In ascending phase, the heights of inflection points in potential temperature profiles are found at 800-850 m and 1300-1350 m. In descending phase, they are located at 950-1000 m and 1450-



1500 m. Whatever the phase considered, the maximum relative humidity is close to saturation
($U$ ~100%) and can even reach supersaturation by 2 to 5 % ($U$ ~102-105%) in descending
phase. This clearly indicates the presence of clouds. Two points need to be underlined. 1) The
heights of the potential temperature inflection points are higher by ~150 m in descending
compared with ascending phases. The landing occurred 70 km further out and 1 h 40 min later
than the launch (Figure 3). This clearly is a fingerprint of both time and space evolution of the
PBL top height around the Concordia station. 2) The presence of a set of two distinct inflection
points, namely two entrainment zones and/or two capping inversion zones where the SLW
clouds develop and/or persist, resemble as if two PBL layers were present above the Concordia
station. The explanation could be that the lowest layer is related to the PBL above Concordia
although the highest layer is either a remnant of the PBL far from Concordia reaching the station
through long-range transport or a fossil layer from the PBL established the day before above
the station. These double layers can be clearly identified on 25 December 2021 at 15:48 UTC
(Figure 5), on 24 January 2022 at 13:51 UTC (Figure S12) and on 28 January 2022 at 06:08
UTC (Figure S13).



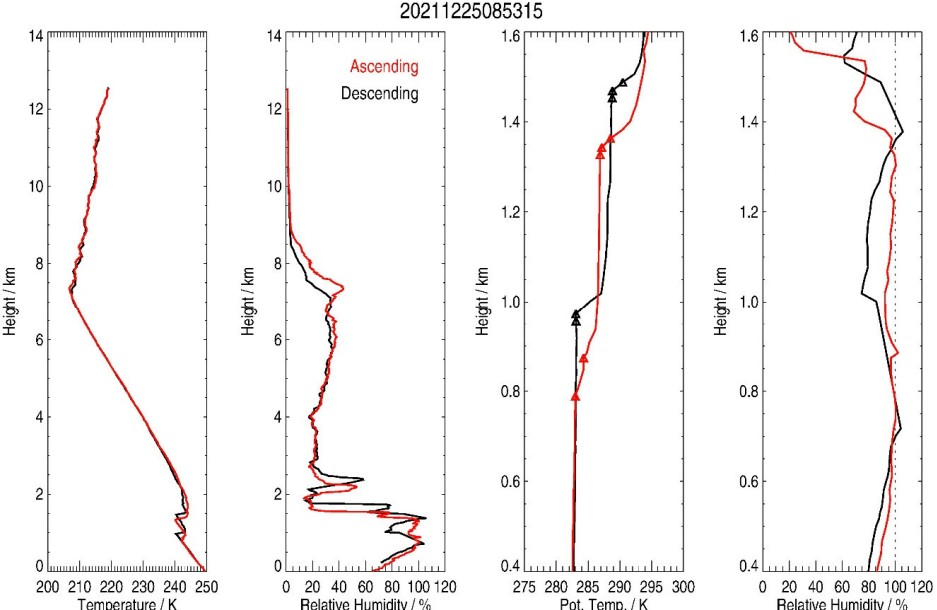


**Figure 4:** (from left to right) Vertical distributions of: temperature (K), relative humidity (%)

observed by the PTU sonde on 25 December 2021 for a launch at 08:53 UTC in ascending (red)

and descending (black) phases over the entire vertical range, and potential temperature (K) and

relative humidity selected from 400 m to 1600 m height. Red and black triangles in the vertical

profiles of potential temperature highlight the presence of inflection points in the ascending and

descending phases, respectively. The vertical dotted line in the right panel indicates the 100%

relative humidity.




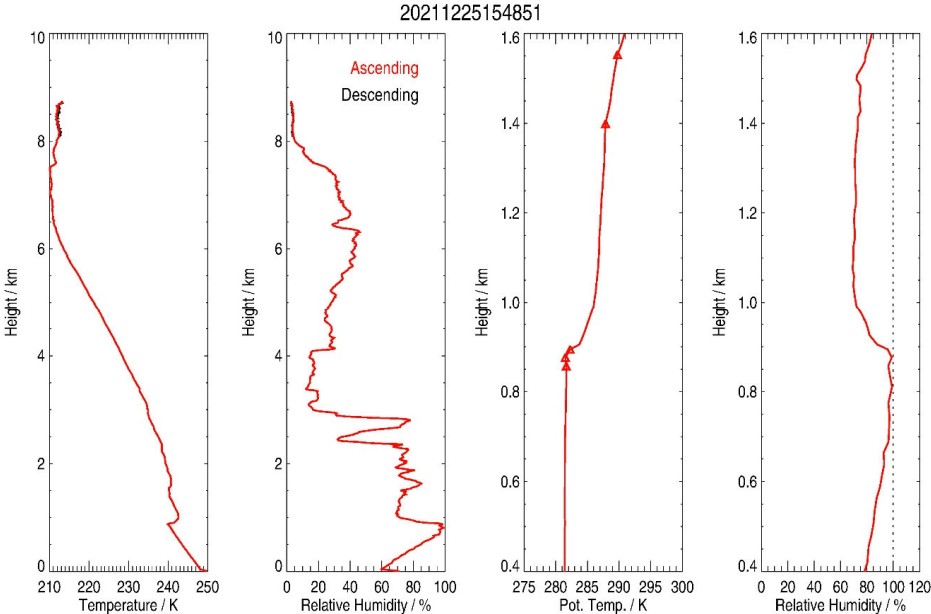


**Figure 5:** (from left to right) Same as Figure 4 but on 25 December 2021 at 15:48 UTC. Note

that, in descending phase (black), only few observations were available after the balloon

reached the ceiling height.


The vertical distributions of $f$, $df/dt$ and SLWC associated with the flights L03 and L04

are shown in Figures 6 and 7, respectively. For both flights, $f$ is rather stable (22.2 and 22.4 Hz,

respectively) along the vertical, with a slight increase between 400 and 600 m during L04. For

L03, the $df/dt$ values are small ($\pm0.001$ Hz s[-1]) except: 1) between 850 and 1000 m (about -

0.005 Hz s[-1]) where an SLW cloud is estimated from 900 to 1000 m with an SLWC of 0.55 g

m[-3] at 950 m and 2) between 1400 and 1500 m (about -0.001 Hz s[-1]) where an SLW cloud is

estimated from 1400 to 1500 m with an SLWC of 0.25 g m[-3] at 1400 m, well above the estimated

1-σ random error of 0.05 g m[-3]. For L04, the $df/dt$ values are small ($\pm0.001$ Hz s[-1]) except: 1)

between 700 and 900 m ($\pm0.005$ Hz s[-1]) where an SLW cloud is estimated from 700 to 900 m

with an SLWC of 0.35 g m[-3] at 850 m and 2) around 1500 m (about -0.001 Hz s[-1]) where an





SLW cloud is estimated around 1500 m with an SLWC of 0.08 g m$^{-3}$, very close to the estimated
1-σ random error of 0.05 g m$^{-3}$. Note that the $df/dt$ values are high below 500 m, reaching
+0.01 Hz s$^{-1}$, but this is not related to the presence of SLW, which translates as negative values
of $df/dt$ (see Equation 1).

For L03 (Figure 6), two sets of potential temperature inflection points are measured at

$H(\theta_{inf})$ = 950-1000 and 1450-1500 m, with no $U$ measurements at these heights. The SLW
clouds derived from the SLWC sonde (900-1000 and 1400-1500 m) are located a within the
lowest part of $H(\theta_{inf})$ and few meters below. For L04 (Figure 7), two to three potential
temperature inflection points are also measured at $H(\theta_{inf})$ = 850-880, 1400 and 1520 m, with
an almost supersaturated atmosphere ($U$ ~100%) at 880 m, and an elevated $U$ ~75% at 1400 m
and $U$ ~80% at 1520 m. The SLW clouds derived from the SLWC sonde (700-900 and 1500
m) are located within the lowest part of $H(\theta_{inf})$ and few meters below, as for the L03 flight.

The SLW cloud heights derived from the SLWC sondes in L03 and L04 are also consistent

with the LIDAR observations (600-800, 800-1000 and 1200-1300 m in L03 and 700-900 m in
L04). Note that, in L04, the SLW cloud layer derived from the SLWC sonde at 1450 m is not
observed by the LIDAR, probably because the underlying SLW cloud at 700-900 m absorbs or
reflects most of the LIDAR radiation, which cannot propagate higher. For L03, the vertically-
integrated in the 900-1000 m layer of the SLWC calculated from the sonde data is about 3.7 g
m$^{-2}$, which falls within the minimum-maximum LWP values observed by HAMSTRAD on that
day (2-6 g m$^{-2}$) whereas, for L04, the SLWC integrated within the 700-900 m layer is 9.0 g m$^{-2}$
slightly larger than the minimum-maximum values observed by HAMSTRAD.



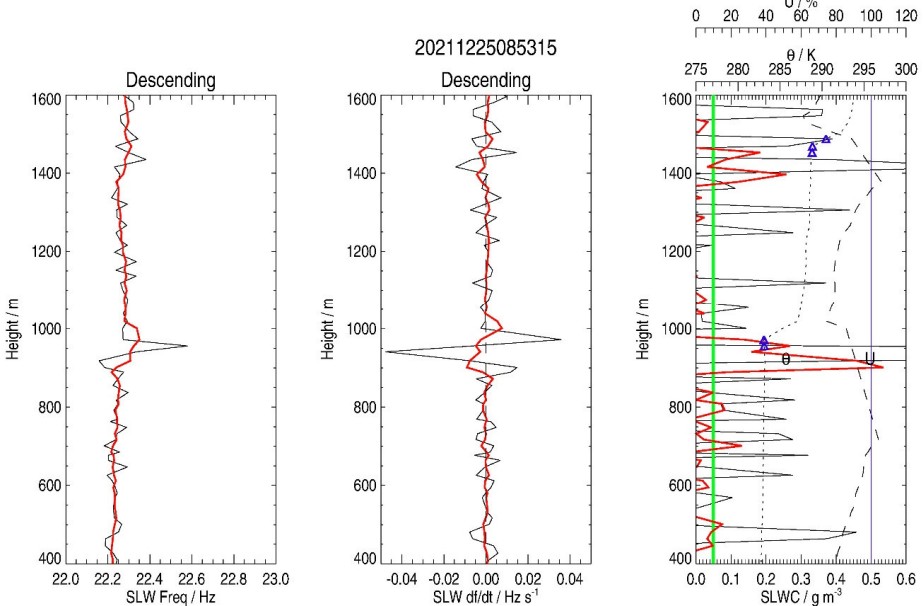


**Figure 6:** Vertical distribution of: (left) SLWC sonde frequency $f$ (black; Hz), (middle) $df/dt$ (black; Hz s$^{-1}$); and (right) sonde-calculated SLWC (black; g m$^{-3}$) on 25 December 2021 at 10:30 UTC (descending phase) for a launch at 08:53:15 UTC. 4-point (20 s) running averages are displayed in red. On the right panel, potential temperature ($\theta$, K) and relative humidity ($U$, %) are shown as dotted and dashed lines, respectively. Blue triangles represent the height of the potential temperature inflection points. The green vertical line represents the estimated one-sigma error (0.05 g m$^{-3}$) of the SLWC calculated from the SLWC sonde observations. The blue vertical line indicates the 100% relative humidity.



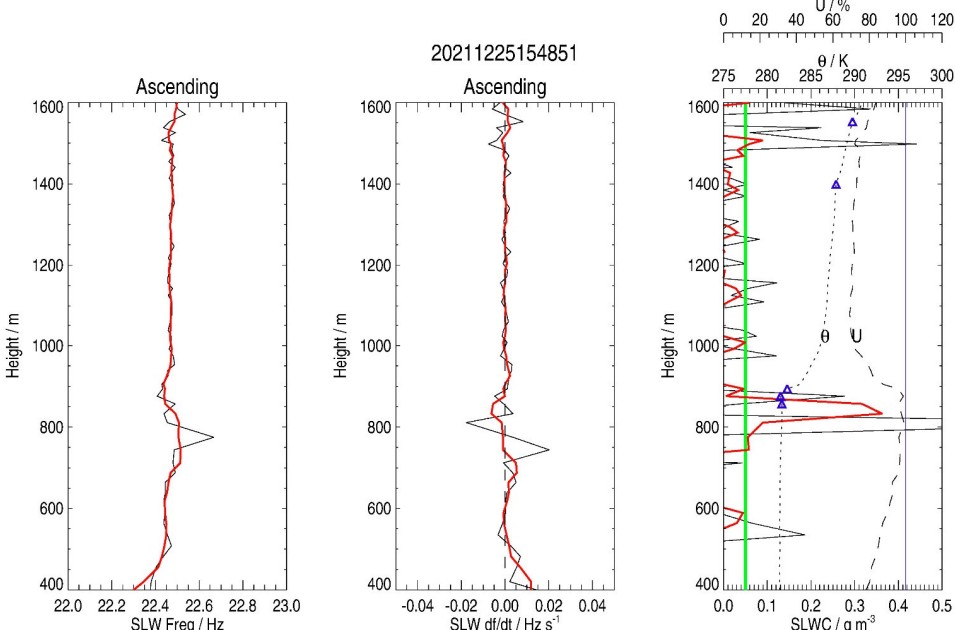

**Figure 7:** Same as Figure 6, but for 25 December 2021 at 15:48 UTC (ascending phase).

*4.3. Launch on 17 January 2022 (cloud-free period)*

The launch on 17 January 2022 at 06:15:15 UTC (L11 in ascending and descending phases) was performed in a cloud-free environment throughout the day, as shown by the LIDAR observations (Figures 8), with associated HAMSTRAD-LWP values of 0.4-1.0 g m$^{-2}$. This launch was an important test to check the behaviour of the SLWC sonde and to quantify the random error associated with the estimation of SLWC. Note that when the sondes reached the ground at the end of the flight, the balloon had travelled a distance of approximately 50 km from the Concordia station after a flight time of 1 h 40 min (Figure 9).



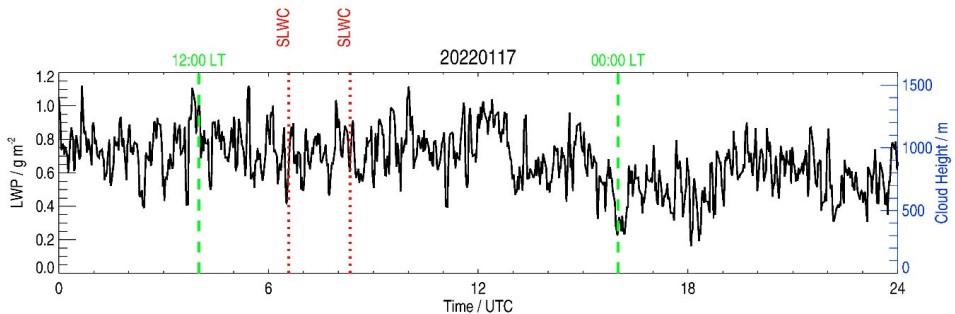


**Figure 8:** Same as Figure 2, but for 17 January 2022, corresponding to a cloud-free condition

period.

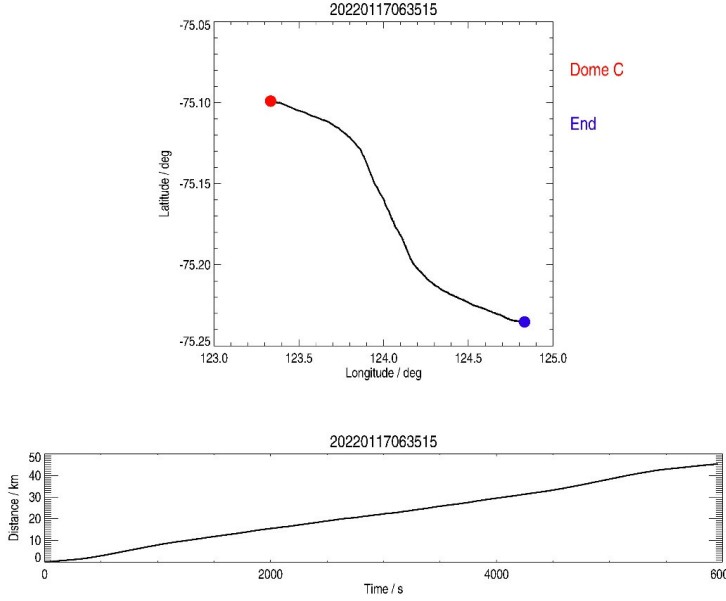


**Figure 9:** Same as Figure 3, but for the meteorological balloon launched on 17 January 2022

at 07:19:05 UTC.


The profiles of $f$, $df/dt$ and SLWC for flight L11 in its ascending and descending phases
are shown in Figures 10 and 11, respectively. $f$ does not vary much along the vertical in both
flight phases with variations lower than ±0.05 Hz producing $df/dt$ values of the order of





$\pm 0.002$ Hz s$^{-1}$. On average, the SLWC oscillates within $\pm 0.05$ g m$^{-3}$. Therefore, we can estimate
the random error in the derived SLWC to be $\sigma = 0.05$ g m$^{-3}$ and conclude that no SLW clouds
were observed with the sonde (although some spikes slightly larger than $\sigma$ are detected at 400
and 1600 m in ascending phase and at 600 and 1200 m in descending phase). This is consistent
with the fact that: 1) the relative humidity is low ($U$ ranging 10-80%), 2) the LIDAR
observations do not show any SLW clouds during the day and 3) the HAMSTRAD LWP is
small (<1.0 g m$^{-2}$).

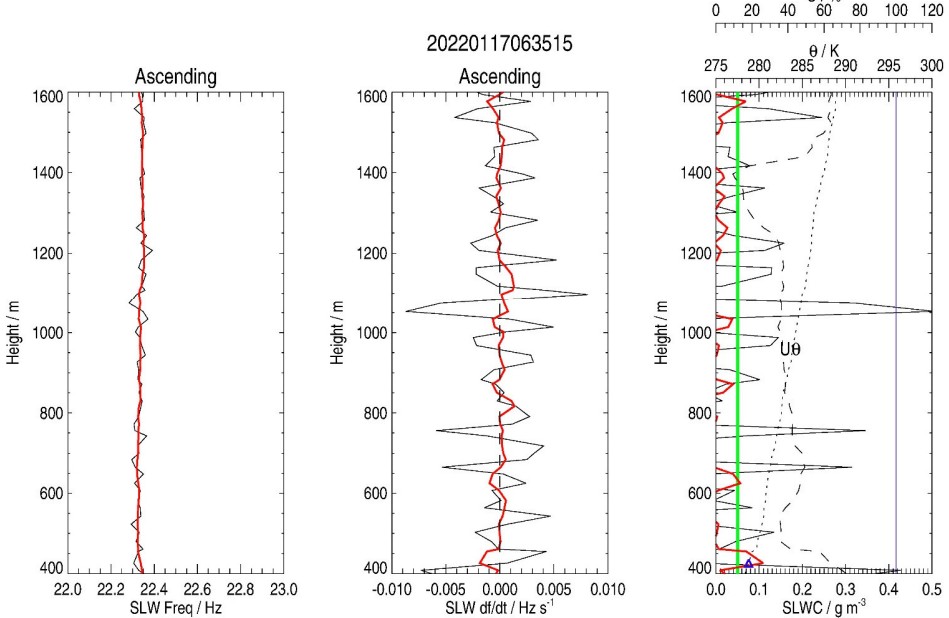


**Figure 10:** Same as Figure 6, but for 17 January 2022 at 06:35 UTC in ascending phase, in a
cloud-free condition.





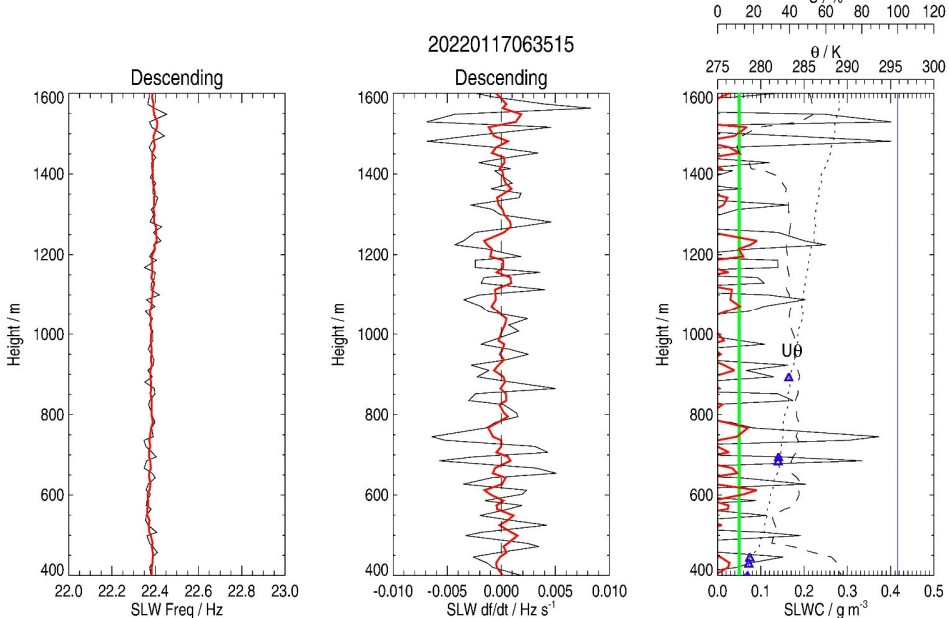

**Figure 11:** Same as Figure 6, but for 17 January 2022 at 06:35 UTC in descending phase, in a cloud-free condition.

*4.4. Analysis of all the other flights*

The first flight (L01) was carried out on 22 December 2021 at 02:24:30 UTC using an unwinder, after the LIDAR detection of an SLW cloud at 400-600 m between 00:00 and 02:00 UTC with an LWP of 8-10.5 g m$^{-2}$ (Figure S1). Unfortunately, just before the launch, the HAMSTRAD-observed LWP decreased to 1.5 g m$^{-2}$, with some remnants of SLW cloud at 500 and 650 m. An SLW cloud is estimated from 400 to 500 m (Figure 12) with an SLWC of 0.35 g m$^{-3}$ at 450 m, well above the estimated 1-$\sigma$ random error of 0.05 g m$^{-3}$. From 400 to 750 m, $U$ increases from 80 to 90% and $H(\theta_{inf})$ is ranging 710-750 m. The LIDAR observed an SLW cloud at 750 m 20 min after launch slightly higher than the cloud height estimated by the SLWC sonde (400-500 m). The integral into the 400-500 m layer of the SLWC measured by the sonde





is about 7.4 g m$^{-2}$, which is within the minimum-maximum values observed by HAMSTRAD
on that day, namely 2-10 g m$^{-2}$.

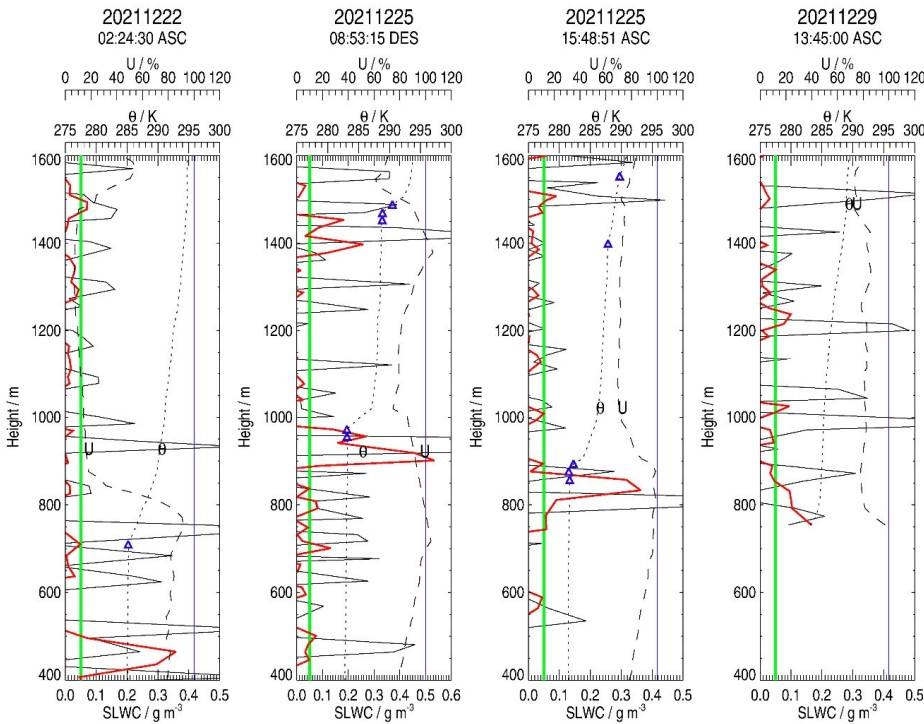


**Figure 12:** (from left to right) Profiles of SLWC (black; g m$^{-3}$) observed on: 22 December 2021
at 02:24 UTC (ascending phase); 25 December 2021 at 10:30 UTC (descending phase) after a
launch at 08:53 UTC; 25 December 2021 at 15:58 UTC (ascending phase) and 29 December
2021 at 13:45 UTC (ascending phase). 4-point (20 s) running averages are displayed in red. The
potential temperature ($\theta$, K) and the relative humidity ($U$, %) are shown as dotted and dashed
lines, respectively. Blue triangles represent the height of the potential temperature inflection
points. The green vertical line represents the estimated one-sigma error (0.05 g m$^{-3}$) of the
SLWC calculated from the SLWC sonde observations. The blue vertical line indicates the 100%
relative humidity.

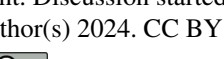



To reduce the duration of instability of the SLWC sonde just after the launch of the balloon,
from 29 December 2021, we no longer used an unwinder but an unwound string of length $L$=40
m (L06 and L07 on 29 December 2021 and L11 on 17 January 2022) or $L$=20 m (L14 on 24
January 2022 and L15 on 28 January 2022). We still used a parachute to make observations
during the descending phase.

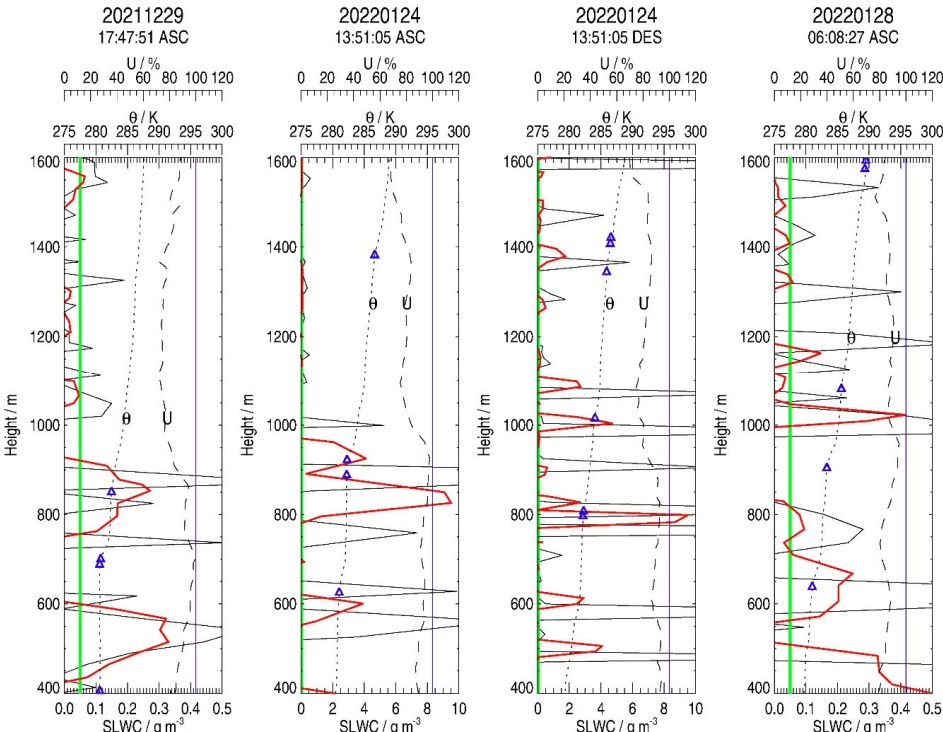


**Figure 13:** (from left to right) Same as Figure 12 but on: 29 December 2021 at 17:47 UTC
(ascending phase); 24 January 2022 at 13:51 UTC (ascending phase); 24 January 2022 at 15:30
UTC (descending phase) after a launch at 13:51 UTC and 28 January 2022 at 06:08 UTC
(ascending phase).

On 29 December 2021, two launches occurred at 13:45:00 UTC (L06 in ascending phase)
and at 17:47:51 UTC (L07 in ascending phase) after more than 2 hours of SLW clouds observed



by the LIDAR (Figure S3) between 600 and 800 m, which continued during the flight. The
launches were associated with HAMSTRAD-LWP values of 1.5-3.0 g m$^{-2}$. Note that, on L06,
the PTU and SLWC sondes only started acquiring data above 750 m in the ascending phase.

On L06 (Figure 12), an SLW cloud is detected between 750 and 850 m with a maximum

of SLWC of 0.16 g m$^{-3}$ and, on L07 (Figure 13), two SLW clouds are estimated, from 400 to
600 m with an SLWC of 0.32 g m$^{-3}$ at 500 m and from 750 to 900 m with an SLWC of 0.28 g
m$^{-3}$ at 850 m. On L06, the potential temperature inflection point is certainly below the height
when the sondes started acquiring (< 750 m) with near-saturated air at 750 m and, on L07, two
potential temperature inflection points are measured at $H(\theta_{inf})$ = 700 and 880 m, with saturated
or near saturated air ($U$ ~100% and ~90%, respectively). The SLW clouds derived from the
SLWC sonde are in the lowermost part or slightly below $H(\theta_{inf})$. The upper SLW cloud
heights inferred from the SLWC sondes in L06 and L07 are also consistent with the LIDAR
observations (600-800 m), but the LIDAR does not detect any cloud between 400 and 600 m.
The amounts of SLWC observed by the sonde and integrated within the layers 750-850 (L06),
400-600 (L07) and 750-900 m (L07) are about 7.5, 33.2 and 23.9 g m$^{-2}$, respectively, slightly
larger (L06) and much larger (L07) than the minimum-maximum values of the LWP observed
by HAMSTRAD on that day (1.0-3.5 g m$^{-2}$). Two important points need to be emphasised to
explain this excess in SLW observed by the sonde on L07. 1) $f$ is not stable along the vertical
during the first few hundred meters after launch (Figure S19), contrary to what was observed
during the previous flights analysed (sections 4.2 and 4.3). And 2) the ascending velocity on
this day was lower ($\omega$ ~3.5 m s$^{-1}$) than the nominal velocity of the air relative to the vibrating
wire (~5 m s$^{-1}$).

On 24 January 2022, we used both the ascending and descending phases of the flight

initiated at 13:51:05 UTC (L14) after more than 2 hours of SLW clouds observed by the LIDAR
(Figure S4) near the surface between 0 and 200 m. In fact, an episode of intense fog developed

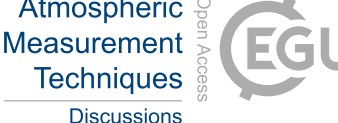

just before the launch. The launch was associated with HAMSTRAD-LWP values of 1.5-3.0 g
$m^{-2}$. One of the main caveats with fog is that, when it is intense, the LIDAR signal cannot
propagate efficiently and the presence of a cloud above the fog layer may not be detected. Note
that, when the sondes reached the ground at the end of the flight, the balloon had travelled a
distance of about 15 km from the Concordia station during 1 h 25 min of flight (Figure S25).
In the ascending phase, two SLW clouds are estimated, around 600 m and from 800 to 1000 m
(Figure 13). Potential temperature inflection points are detected at $H(\theta_{inf}) = 630$ and 920 m,
with air close to saturation ($U$ ~90-95%) and, to a lesser extent, at 1400 m. In the descending
phase, several spikes of SLW clouds were detected below 1200 m, but the two most intense
were located around 800 and 1000 m (Figure 13). The potential temperature inflection points
were measured at $H(\theta_{inf}) = 800$, 1020, 1340 and 1420 m, with relative humidity $U$ ranging 85-
95%. In both phases, the SLW clouds derived from the SLWC sonde are located in the
lowermost part of the entrainment/capping inversion zone. During the flight, the LIDAR
measured two SLW clouds around 350 and 800 m, in addition to near-surface fog. This means
that the SLW cloud around 800 m was detected by all the instruments, while an underlying
SLW cloud was detected around 600 m by the sondes and much lower (at 350 m) by the LIDAR,
slightly below the lowest level where the SLWC sondes start to work well. The SLWCs
observed by the sonde and integrated within the layers 800-1000 m (ascending phase) and 750-
850 m (descending phase) are about 575.3 and 18.9 g $m^{-2}$, respectively, much larger than the
minimum-maximum values observed by HAMSTRAD on that day (1-5 g $m^{-2}$). Two important
points must be emphasized in order to explain this excess in SLW by the sonde. 1) As far as the
flight L14 is concerned, $f$ was not stable along the vertical during the first few hundred meters
after launch (Figures S21 and S22), contrary to what was observed during the previous flights
analysed (sections 4.2 and 4.3). Above all, the flight was carried out when a fog episode
developed over the station. Some ice crystals and/or SLW droplets could well have adhered to





the wire of the SLWC sonde before the launch and perturbed the nominal operation of the sonde
system, namely the vibration frequency and the post-launch stabilization process.

The last launch of the summer campaign was performed on 28 January 2022 at 06:08:27

UTC (L15 in ascending phase) after more than 2 hours of SLW clouds observed by the LIDAR
(Figure S5) at 600-800 m and 900-1000 m. The launch was associated with HAMSTRAD-LWP
values of 3.0-3.5 g m$^{-2}$. After the launch, the LIDAR detected SLW clouds at about 1000 m.
Excluding the large signal at 400 m which is probably due to some residual vibrations from the
launch (Figure 13), two SLW clouds are estimated from 600 to 800 m with an SLWC of 0.25 g
m$^{-3}$ at 650 m and around 1000 m with an SLWC of 0.40 g m$^{-3}$. Three potential temperature
inflection points are estimated at $H(\theta_{inf}) = 650$, 910 and 1080 m, with $U$ ranging 85-95%. The
SLW clouds detected by the SLWC sonde (at 650 and 1000 m) are well within the
entrainment/capping inversion zone and at heights consistent with the LIDAR observations
(600-800 and 1000 m). The SLWC observed by the sonde and integrated within the 600-800 m
and the 950-1050 m layers are about 10.1 and 7.3 g m$^{-2}$, respectively, slightly larger than the
minimum-maximum values observed by HAMSTRAD on that day (2-5 g m$^{-2}$).

**5. Synthesis and Discussion**
*5.1. SLW cloud*

Our study reveals that, during the 2021-22 summer campaign at Concordia, the detection

of the SLW cloud heights shows high agreement between the remote sensing observations with
the LIDAR and the in-situ observations with the SLWC sondes. The clouds are generally
located just below the height $H(\theta_{inf})$ of an inflection point in the potential temperature profile,
within a layer where the relative humidity $U$ exceeds 80%, sometimes reaching saturation
(100%) and in the lowermost part of the entrainment/capping inversion zone depending on the
local time. These results are in agreement both with the theory of the diurnal evolution of the



planetary boundary layer (PBL), for which boundary-layer clouds develop at the top of the PBL
(Stull, 2012), as well as with the first studies carried out at Concordia based only on remote
sensing observations (Ricaud et al., 2020). The presence of the SLW clouds is also observed 1)
below the height of the inflection point in potential temperature profile during the High-
performance Instrumented Airborne Platform for Environmental Research (HIAPER) Pole-to-
Pole Observations global transects over the Southern Ocean (Chubb et al., 2013) and 2) around
the height of the inflection point in temperature profile above the South Pole station from
backscatter LIDAR signal (Lawson and Gettelman, 2014).
The SLWC maxima measured by the sondes were ranging 0.2-0.5 g m$^{-3}$ in nominal
operations. This is consistent with: 1) the observations performed in the Arctic with the same
sondes and with a surface-based AMF3 microwave radiometer (maxima of 0.3-0.4 g m$^{-3}$)
attached to a tethered balloon (Dexheimer et al., 2019), 2) in situ airborne observations from
HIAPPER (maximum of 0.47 g m$^{-3}$) (Chubb et al., 2013), 3) the 580-s observations from the
Southern Ocean Clouds, Radiation, Aerosol Transport Experimental Study (SOCRATES)
airborne campaign over the Southern Ocean (maximum of SLWCs of 0.60 g m$^{-3}$) and 4) results
from three climate models (maxima of SLWCs ranging from 0.36 to 0.40 g m$^{-3}$) (Yang et al.,

2021).

It should also be noted that the variations at scales smaller than 100 m in the vertical
profiles of the SLWC observations are smoothed out because of: 1) the 5-s integration time of
the raw measurements, 2) the method of deriving the SLWC from equation (1) which requires
the use of the vertical derivative of $f$, and 3) the 4-point running average applied to the
observations to minimise the effect of large signal frequency undulations on the retrieved
SLWC. Therefore, the actual location of the SLW clouds from the SLWC sondes might be
slightly displaced compared with the actual location of the entrainment/capping inversion zone
derived from the PTU sondes.





*5.2. Vertically-integrated SLWCs*
The vertically-integrated SLWCs calculated from in-situ observations were consistent with
the minimum-maximum LWPs observed by HAMSTRAD (flights L01 and L03 with
unwinders) or slightly larger than the maximum of LWP (flights L04 and L15 with unwinder
and a fixed string of $L = 20$ m, respectively). Flight L07 (fixed string of $L = 40$ m) gave a
vertically-integrated SLWC greater than that observed by HAMSTRAD by a factor of 5-10,
and we can point out that the ascent vertical velocity was certainly too low for the sonde to
operate nominally. Finally, for the flight carried out when a fog episode was present (L14), the
vibrating wire of the SLWC sonde was probably affected by this event before launch producing
an unrealistically large amount of SLWC during the flight. Furthermore, our best results were
obtained with an SLWC sonde attached to the balloon with an unwinder.
In nominal operations, LWPs from the sondes were consistent with HAMSTRAD
observations (1-14 g m$^{-2}$). Nevertheless, LWPs observed over Concordia deep inside the
Antarctic Plateau were much less than those observed in the Arctic (15-40 g m$^{-2}$ in Dexheimer
et al. (2019) and greater than 50 g m$^{-2}$ in Zhang et al. (2019)) and over the coastal Antarctic
station of McMurdo (10-50 g m$^{-2}$ in Zhang et al. (2019) and 40-60 g m$^{-2}$ in Hines et al. (2021)).
*5.3. Quality/sensitivity of the SLWC sonde*
Flying during a cloud-free period helped to characterize the random RMSE $\sigma$ associated
with the retrieved SLWC. Compared to the other flights carried out during the campaign, the
cloud-free flight (L11 with a fixed string of $L = 40$ m) was nominal with a low variability of $f$
during the ascent and descent phases for heights above 400 m, from which we estimated that $\sigma$
was about 0.05 g m$^{-3}$.
The way the balloon is released is a key issue for the stability of the SLWC sonde and
needs to be addressed whenever the SLW clouds of interest are near the surface within the PBL.
Irrespective of the method used (unwinder or unwound string), during the 2021-22 summer



575 season we were unable to find a way to stabilise the sonde in less than 60 s after launch. One

576 of the main difficulties was that some SLW clouds were located around $400 \pm 100$ m and, in

577 this case, we were unable to determine whether the variations in the frequency derivatives were

578 due to an instability of the sonde or to a real SLW cloud.

579 Finally, in our opinion, the optimum way to launch the SLWC sonde was to attach it to the

580 balloon with an unwinder although we obtained one scientifically-exploitable flight using an

581 unwound string of length $L$=40 m (L07 on 29 December 2021). However, we have only 9 flights

582 and more flights would be needed to confirm this. We have already highlighted the difficulty

583 of numerical weather prediction models to reproduce the SLW clouds over Concordia, which

584 produces erroneous cloud radiative forcings (Ricaud et al., 2020) along with biased temperature

585 and humidity profiles in the PBL (Ricaud et al., 2023). Therefore, in situ observations, although

586 difficult to deploy, still remain a key tool for improving NWPs in these harsh environments.

588 **6. Conclusions**

589 The present study intended to observe in situ SLW clouds above the Concordia station by

590 means of sondes sensitive to SLW especially developed by the Anasphere Company. These

591 sondes were attached to meteorological balloons and connected to standard Vaisala PTU sondes

592 during the 2021-2022 summer campaign. These launches were coupled with observations from

593 a backscattered LIDAR providing the nature and height of the clouds, and a microwave

594 radiometer providing the LWP. Over a total of 15 launches, 7 were scientifically exploitable

595 mainly above 400 m agl, a threshold height imposed by the time the SLWC takes to stabilize

596 after the launch.

597 In general, during nominal operations, the SLWC sondes detected SLW clouds in a vertical

598 domain consistent with LIDAR observations, and the LWP values either obtained by

599 HAMSTRAD or vertically-integrated from the SLWC values calculated from the sonde


observations were consistent in spite of their low values ($< 10$ g m$^{-2}$). Unfortunately, on some
occasions far from nominal operation (surface fog, low vertical ascent of the balloon), the sonde
vertically-integrated SLWCs were overestimated by a factor of 5-10 compared to the
HAMSTRAD LWPs.
Although the vertical resolution of the SLWC observations is around 100 m due to the
methodology employed (4-point running average of 5-s integration time) and the vertical ascent
of 5 m s$^{-1}$, the SLW clouds were observed in a layer close to saturation ($U > 80\%$) or saturated
($U \sim 100\text{-}105\%$) just below or at the lowermost part of the entrainment zone or capping inversion
zone which exists at the top of the PBL and is characterized by an inflection point in the
potential temperature vertical profiles. Consequently, our results are consistent with the
theoretical view that SLW clouds form and pertain at the top of the PBL.
Because of the positive scientific results obtained during this first balloon campaign and
since the second campaign in 2022-2023 was technologically successful using a VTOL drone,
we forecast a new summer campaign to probe the PBL with an SLWC sonde aboard a drone.
The main advantages of the drone compared with the meteorological balloon are that: 1) it can
fly every day or even twice a day with the same SLWC sonde onboard minimizing the number
of SLWC and PTU sondes used and 2) it does not require Helium gas that is coming to be more
and more difficult and costly with time.

**Data availability**
HAMSTRAD data are available at http://www.cnrm.meteo.fr/spip.php?article961&lang=en
(Ricaud, 2024). The tropospheric depolarization LIDAR data are reachable at
http://lidarmax.altervista.org/lidar/home.php (Del Guasta, 2024). Radiosondes are available at
http://www.climantartide.it/dataaccess/RDS_CONCORDIA/index.php?lang=en     (Grigioni,

2024).






**Author contribution**

PR, MDG, GC, AR, PG and JB provided the observational data. PR developed the methodology with the help of all co-authors. All the co-authors participated in the data analysis and in the data interpretation. PR prepared the manuscript with contributions from all co-authors.

**Competing interests**

The authors declare that they have no conflict of interest.

**Acknowledgements**

The present research project Water Budget over Dome C (H2O-DC) has been approved by the Year of Polar Prediction (YOPP) international committee. The permanently staffed Concordia station is jointly operated by Institut polaire français Paul-Emile Victor (IPEV) and the Italian Programma Nazionale Ricerche in Antartide (PNRA). The tropospheric LIDAR operates at Dome C from 2008 within the framework of several Italian national (PNRA) projects. We would like to thank all the winterover personnel who worked at Dome C on the different projects.

**Financial support**

The HAMSTRAD programme 910 and the SLW-CLOUDS programme 1247 were supported by IPEV, the Institut National des Sciences de l'Univers (INSU)/Centre National de la Recherche Scientifique (CNRS), Météo-France, and the Centre National d'Etudes Spatiales (CNES).




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
