# Peer review of "In situ observations of supercooled liquid water clouds over Dome"

_Atmospheric Measurement Techniques, 2024_

## Author Response (AR1)

**Version 03, 1 July 2024**

**Manuscript Title:** *In situ observations of supercooled liquid water clouds over Dome C, Antarctica by balloon-borne sondes* **by Ricaud et al.**

**RESPONSES TO THE EDITOR**

$\rightarrow$ Dear Editor.

Specific changes have been made in response to the reviewers' comments and are described below. The reviewers' comments are presented in blue.

We have acknowledged the two anonymous reviewers. One sentence has been inserted in the Acknowledgements.

**Referee #1**

→ Specific changes have been made in response to the reviewer's comments and are described below. The reviewer's comments are recalled in blue.

The manuscript describes the measurement of cloud supercooled liquid using in situ measurements at the Concordia station in Antarctica. Cloud supercooled liquid water is an important parameter that needs to be better measured, especially in Antarctica, because it is not well described in numerical models. In addition, measurements in Antarctica are always very challenging. Therefore, the manuscript has an important contribution to the readers of AMT. However, the manuscript has some major problems, which I comment on below. I proposed to accept the manuscript after a major revision.

→ Thank you for your positive feedback on the manuscript.

**Major Comments**

→ We have split the paragraph containing the Referee's major comments into several points, which we will deal with one after the other.

The manuscript spends several pages discussing the potential temperature, the flight description, and the radiosonde trajectories and misses the most important part, the measurement of cloud liquid water. I suggest the authors focus the manuscript on the main points.

→ We have enlarged the discussion on the SLW cloud observations. Potential temperature and radiosonde trajectories are nevertheless key points (among other ones) to interpret the measurements of the SLW clouds.

I miss a statistical calculation of the errors between the different instruments.

→ We have inserted these errors in the revised version of the manuscript within Table 2. Biases and RMSEs are better characterized from observations in cloud-free conditions. To sum up, for SLWC sondes, we have found a bias and a RMSE in SLW content (SLWC) of 0 and 0.08 g m$^{-3}$, respectively, and, for HAMSTRAD, we have found a bias and a RMSE in liquid water path (LWP) of 1.0 and 0.2 g m$^{-2}$, respectively.

We have also inserted a new Table 3 that describes for each instrument used in our analysis (HAMSTRAD, LIDAR and the SLWC sonde) the viewing geometry and comments on the information on 1) the sensitivity of the cloud type (overcast and/or scattered) that can be sounded, 2) the heights of the clouds, 3) which parameter (SLWC and/or LWP) is observed, and 4) any other information relevant to the study.

**Table 3.** Description of the viewing geometry and comments relative to each instrument used in our analysis: HAMSTRAD, LIDAR and SLWC sonde.

| Instruments | Viewing Geometry | Comments |
|---|---|---|
| HAMSTRAD | 0-90° zenithal angle, to the East | - Overcast
- Only LWP is measured
- No information on the SLW cloud height |

| | | - Continuous and automated observations |
|---|---|---|
| LIDAR | 0.4° off-zenith | - Scattered and overcast
- Information on the SLW cloud height
- No information on either SLWC or LWP
- Presence of precipitation and/or low clouds can alter the observations of clouds above
- Continuous and automated observations |
| SLWC sonde | In-situ | - Increasing horizontal distance from the station as the flight progresses
- Scattered and overcast
- Information on the SLW cloud height
- SLWC is measured along the vertical and LWP can be inferred
- Sporadic and manual observations |

I would like to see how all measurements and the selected ones compare with remote sensing observations. I would like to see the LIDAR measurement profiles combined with the sonde calculated SLWC in Figures 6, 10, 11 and 12.

→ This is a very good point.

We have redrawn Figures 6, 7, 12 and 13 (Figs. 10 & 11 are for cloud-free conditions) and Figures S15-S24 by adding the vertical extension of the SLW clouds observed by the LIDAR in a ±1-hour time window centered on the launch time of the balloon (ascending phase) and/or on the time of the flight end (descending phase) in yellow or orange, respectively. As an example, we show below the Figures 6, 12 and 13 of the revised manuscript.

[Figure]

**Figure 6:** Vertical profiles of: (left) SLW sonde frequency f (black; Hz), (middle) df/dt (black; Hz s$^{-1}$); and (right) sonde-calculated SLWC (black; g m$^{-3}$) on 25 December 2021 at 10:30 UTC (descending phase for a launch at 08:53:15 UTC). 4-point (20 s) running averages are displayed in red. On the right panel, potential temperature ($\theta$, K) and relative humidity (U, %) are shown as dotted and dashed lines, respectively. Blue triangles represent the height of the potential temperature inflection points. The green vertical line represents the estimated one-sigma error (0.08 g m$^{-3}$) of the SLWC calculated from the SLWC sonde observations. The blue vertical line indicates the 100% relative humidity. The vertical extensions of the SLW clouds as observed by the LIDAR in a ±1-hour time window centered on the launch time (ascending phase) or on the time of the flight end (descending phase) are highlighted in yellow or orange, respectively.

[Figure]

**Figure 12:** (from left to right) Profiles of SLWC (black; g m$^{-3}$) observed on: 22 December 2021 at 02:24 UTC (ascending phase); 25 December 2021 at 10:30 UTC (descending phase after a launch at 08:53 UTC); 25 December 2021 at 15:58 UTC (ascending phase) and 29 December 2021 at 13:45 UTC (ascending phase). 4-point (20 s) running averages are displayed in red. The potential temperature ($\theta$, K) and the relative humidity (U, %) are shown as dotted and dashed lines, respectively. Blue triangles represent the height of the potential temperature inflection points. The green vertical line represents the estimated one-sigma error (0.08 g m$^{-3}$) of the SLWC calculated from the SLWC sonde observations. The blue vertical line indicates the 100% relative humidity. The vertical extensions of the SLW clouds as observed by the LIDAR in a ±1-hour time window centered on the launch time (ascending phase) or on the time of the flight end (descending phase) are highlighted in yellow or orange, respectively.

[Figure]

**Figure 13:** (from left to right) Same as Figure 12 but on: 29 December 2021 at 17:47 UTC (ascending phase); 24 January 2022 at 13:51 UTC (ascending phase); 24 January 2022 at 15:30 UTC (descending phase) after a launch at 13:51 UTC and 28 January 2022 at 06:08 UTC (ascending phase).

One point that is not clear to me is the multilayer of supercooled liquid water clouds in different layers. In my interpretation, I would expect only one layer, as shown by the LIDAR in Figure 2. Is the LIDAR attenuated for the other layers? I suspect not, these are very thin clouds.

→ You are right to raise this issue. The SLW clouds are created at the top of the boundary layer within the entrainment zone, and can pertain at this height below the capping inversion zone even after the collapse of the convective boundary layer (Stull, 2012; and Figure 18 in Ricaud et al., 2020). However, we have already observed two layers of SLW clouds above Concordia station (see Fig. R01; Ricaud et al., *manuscript in preparation*). Our understanding/explanation is the following:

1) When the PBL collapses (transition between a mixed layer and a stable layer), the top of the PBL decreases rapidly towards the surface. Within these stable conditions, SLW clouds and/or liquid fog can form very close to the surface while, just below the remnant of the capping inversion zone, the SLW cloud still persists. Consequently, we may have 2 SLW cloud layers.

2) Independently of the diurnal variation of the PBL locally at Concordia and thus, of the presence of SLW clouds at the top of this local PBL, we may well have SLW clouds created elsewhere over the continent, at a height that can be different from the PBL top height at

Concordia. These SLW clouds created far away can then be advected to Concordia via transport processes, such as isentropic transport. This will thus produce 2 layers of SLW clouds. In that case, we can say that these later SLW clouds still pertain in a "fossil PBL" originated away from the Concordia station. An illustration is given in the Figure R01 below.

3) Very locally, we have already observed, as e.g. on 24 January 2022 (Fig. S1), liquid fog close to the surface within a layer from the ground to about 200 m height. It is not clear, at the present time, whether these liquid fog phenomena are originated from the human activity at the station.

To sum up, we may well have at the same time over Concordia station several SLW clouds within different layers. But, whatever the process highlighted (1-3), it is interesting to note that the double-layer SLW clouds are always present slightly below or in the vicinity of the height of the inflection points in the potential temperature profile.

[Figure]

Figure R01: (Top) On 23 January 2023, diurnal variation of: the layers with a relative humidity greater than 83 % (blue areas) and isentropes (black lines) calculated from the temperatures observed every 3 hours by balloon-borne sondes from 00:00 to 24:00 UTC; SLW clouds (red areas) and precipitation (grey areas) as measured by the LIDAR; PBL top height as calculated by the numerical weather prediction (NWP) model ARPEGE (purple line) and ERA5 reanalyses (yellow line). Inflection points in the potential temperature profiles are shown with black triangles. (Bottom) Video capture of the sky with the camera installed on top of the Physics Shelter at (a) 01:00, (b) 06:00, (c) 10:50, (d) 17:00, (e) 21:00 and (f) 23:00 UTC showing either clear sky (b, d, f) or SLW cloud (a, c, e) conditions (Ricaud et al., *manuscript in preparation*).

Regarding the LIDAR observations, it is clear that the upward LIDAR signal cannot efficiently propagate through a cloud and, if thick, cannot even penetrate the cloud. The SLW clouds investigated in our study although thin clouds can contain ice particles (mixed-phase clouds) and/or precipitation (ice crystals falling down to the surface). It is the reason why we have shown in the revised version 1) the SLW clouds diagnosed from the LIDAR depolarization ratio, but also 2) the profiles of the depolarization ratio that shows up both ice clouds and

precipitation and 3) the backscattered signal. For example, Fig. 2 below highlights two interesting points. 1) Slightly above the SLW clouds (on 06:00-10:00 UTC and 13:00-18:00 UTC) is associated a vertical domain where the depolarization reaches 15% (ice crystals) and the backscatter signal drops. This means that any clouds above the SLW clouds will have almost no chance to be sounded by the LIDAR instrument since the attenuation of the signal is too large. As a consequence, while the SLWC sonde is able to detect a multi-layer SLW cloud, the LIDAR can show only one layer (the lowermost). This is also true when a thick liquid fog is present over the station as e.g. on 24 January 2022 (see for instance Fig. S4).

[Figure]

**Figure 2:** Diurnal variation on 25 December 2021 (UTC Time) along the vertical of: (Top) the profile of the LIDAR backscatter signal (A.U., Arbitrary Unit); (Center) the profile of the LIDAR depolarization ratio (%); (Bottom) the Liquid Water Path (LWP) measured by HAMSTRAD (g m$^{-2}$, black solid line) superimposed with the SLW cloud thickness (red area) derived from the LIDAR observations (red y-axis on the right). Two vertical green dashed lines indicate 12:00 and 00:00 LT. The thick red vertical dashed lines indicate the time when balloon observations with SLWC sondes were performed in ascending (ASC) or descending (DES) phase while the thin red vertical solid line indicates the launch time of the balloon for which observations in the descending phase were performed.

*My main suggestion is to avoid all this parallel discussion in the manuscript and focus on the main points - how these measurements compare with remote sensing. Do the measurements make sense? What is the bias, the mean square error?*

→ These points were detailed in the previous sections of our responses.

*Are these measurements relevant for calculating an accurate radiative effect of these clouds compared to just ice clouds? How many W.m-2 is the error, is it larger than the estimated 40 W.m-2, what is the impact on radiative transfer if this multilayer is considered at the wrong height?*

→ It is beyond the scope of the present paper to consider the SLW cloud radiative forcing. We have already studied this point and presented the SLW cloud radiative forcing for 2 case studies (Ricaud et al., 2020) and on a climatological point of view (Ricaud et al., 2024). We already noticed that the cloud radiative forcing is greater for SLW cloud than for ice cloud at Dome C. NWP models and reanalyses (whatever their vertical and horizontal resolutions) favor ice clouds at the expense of SLW clouds. Based on a climatological study covering December 2018-2021 over Concordia station, we have found that the SLW cloud radiative forcing calculated with ERA5 reanalyses presented a deficit of 15 W m$^{-2}$ with respect to observations (see Figure R02; Ricaud et al., *manuscript in preparation*).

[Figure]

**Figure R02:** Liquid Water Path (LWP, g m$^{-2}$) vs. SLW Cloud Radiative Forcing (W m$^{-2}$) using ERA5 data over Concordia station for different time windows in December: 1943-1952 (black), 1983-1992 (blue), 2013-2022 (green) and 2018-2021 (red). A fit through observations performed in December 2018-2021 (Ricaud et al., 2024) is highlighted in orange (Ricaud et al., *manuscript in preparation*).

Furthermore, the radiative forcing evaluated from the multilayer SLW clouds and/or the actual height of the SLW clouds is worthwhile mentioning but requires a tremendous effort of modelling. This is again far away from the scope of the present paper.

Ricaud, P., Del Guasta, M., Bazile, E., Azouz, N., Lupi, A., Durand, P., Attié, J.-L., Veron, D., Guidard, V., and Grigioni, P. : Supercooled liquid water cloud observed, analysed, and modelled at the top of the planetary boundary layer above Dome C, Antarctica, Atmos. Chem. Phys., 20, 4167–4191, https://doi.org/10.5194/acp-20-4167-2020, 2020.

Ricaud, P., Del Guasta, M., Lupi, A., Roehrig, R., Bazile, E., Durand, P., Attié, J.-L., Nicosia, A., and Grigioni, P. : Supercooled liquid water clouds observed over Dome C, Antarctica : temperature sensitivity and cloud radiative forcing, Atmos. Chem. Phys., 24, 613–630, https://doi.org/10.5194/acp-24-613-2024, 2024.

What are the limitations of this method?

→ Based on Table 3, we can state that there is no single method to optimally characterize the SLW clouds above the Concordia station. The three instruments we have used show pros and cons, provide different information on the SLW cloud (height-resolved or not, LWP observation or not, zenith-viewing or angularly-integrated, in-situ observation but moving away from the station). During the summer 2022-23 campaign, we have also used a drone with SLWC sondes onboard to observe in situ the SLW clouds. Unfortunately, no SLW clouds persisted for a long enough period to allow a flight to be made (Ricaud et al., 2023). A new proposal has therefore been submitted to the French polar Institute in order to sound SLW clouds using drones.

Ricaud, P., Medina, P., Durand, P., Attié, J.L., Bazile, E., Grigioni, P., Del Guasta, M. and Pauly, B.: In Situ VTOL Drone-Borne Observations of Temperature and Relative Humidity over Dome C, Antarctica. Drones, 7(8), 532; https://doi.org/10.3390/drones7080532, 2023.

How to avoid such out-of-scale values as in Flight 14?

→ As stated in the manuscript, our explanation for getting out-of-scale values as in flight L14 is that the SLWC sondes were not launched in a cloud-free environment. Liquid fog was present near the surface (see Fig. S4) and the vibration frequency of the wire was thus modified since liquid droplets already transformed into ice at the surface of the wire. It is the only day (period) when such out-of-scale values appeared.

Why is the LIDAR liquid water cloud so different from the measured one?

→ On 24 January 2022, the LIDAR does show some liquid clouds (or liquid fog) in the very lowest layers (below 200-300 m). Therefore, on that day, the laser beam has probably not been able to penetrate within the cloud and sound the atmosphere above the lowermost cloud. It is the reason why, apparently, the LIDAR did not detect any SLW cloud above 200-300 m height.

Instead of only showing the presence of SLW clouds from LIDAR (see e.g. Figs. 2 or S1 in the first version of the manuscript or in the Supplementary material, respectively), we now show systematically the backscatter signal and the depolarization ratio of the LIDAR (new), together with the LWP from HAMSTRAD and the vertical distribution of the SLW clouds (if any) from the LIDAR (see replies to the comments before).

What is the impact on the radiation simulation?

→ It is beyond the scope of the present paper to evaluate the SLW cloud radiative forcing because we already focused on this subject in two different papers. The conclusions are that weather and climate models have difficulty in simulating these SLW clouds and hence to evaluate their radiative forcing. As presented before, Figure R02 shows that ERA5 reanalyses

exhibit a systematic 15 W m$^{-2}$ deficit in SLW radiative forcing compared to the observations above Concordia station (Ricaud et al., *manuscript in preparation*).

**Minors Comments**

Line 58 – Please add references

→ Done. New reference inserted. We have added:

"(see e.g. Fogt and Bromwich, 2008)"

Fogt, R.L. and Bromwich, D.H.: Atmospheric moisture and cloud cover characteristics forecast by AMPS. *Weather and forecasting*, *23*(5), 914-930, 2008.

Line 61 – Please add references

→ Done. New reference inserted. We have added:

"(see e.g. Ricaud et al., 2020)"

Line 68 – LWP and SLWC are the same, so why call them by different names?

→ No, LWP and SLWC are different variables. SLWC represents the Supercooled Liquid Water Content in unit of g m$^{-3}$ while LWP represents the Liquid Water Path in unit of g m$^{-2}$, namely the total column cloud liquid water i.e. the vertical integral of SLWC.

Line 88 – "At Concordia station, several studies from remote-sensed observations already took place to evaluate....". "4) the differences between observations and model simulations of SLW clouds." This is not done at the Concordia station, and remote sensing is not used. Please write this paragraph without using numbers but a coherent description of the studies using Concordia data.

→ In the item "4) the differences between observations and model simulations of SLW clouds.", we refer to remote sensing observations with LIDAR and HAMSTRAD radiometer, both installed at Concordia, and modelling from the ARPEGE NWP (see Ricaud et al., 2020). Nowadays the climatological study (see e.g. Fig. R02) is actually being undertaken in Concordia.

We have rewritten the paragraph by inserting the appropriate references to Ricaud et al. (2020) for the item 4) and Ricaud et al. (2024) for the items 5) and 6).

Line 131—Please clarify. When the SLW reaches the wire, I would expect the liquid to be instantly converted to ice.

→ It is probably the Line 141 (instead of L. 131). Yes of course, it is the principle of the measurement technique. We have clarified this point to avoid any ambiguity.

> The Anasphere's vibrating-wire sonde records a vibrating wire's frequency as ice accumulates along its length (Serke et al., 2014). When the SLW reaches the wire, liquid droplets are instantly converted into ice. These frequency measurements,

combined with collocated meteorological measurements, can be used to determine the SLWC of the surrounding air. The SLWC sonde actually measures the frequency of the vibrating wire. Since this frequency $f$ varies according to the change in mass of the wire, its derivative with respect to time $df/dt$ can be used to calculate the SLWC collected by the wire.

Line 136 - its derivative with respect to (wrt) time - the abbreviation wrt is standard jargon in mathematical papers - please avoid it.

$\rightarrow$ Done, both in the manuscript and in the supplementary materials.

Line 142 - the vertical velocity is variable and could be updraft or downdraft. Please clarify.

$\rightarrow$ ω is the velocity of air relative to the wire, irrespective of its direction (upward, downward, etc.). During the ascending phase, given that the balloon has upward buoyancy, it always rises with respect to the air parcel it is in. The nominal operation of the SLWC sonde requires that it is well working with an air flow of about 5 m s$^{-1}$. It is the reason why the balloon pressure is set up for an average ascending rate (with respect to the ground) of ~5 m s$^{-1}$. During the descending phase, after the balloon has burst, the sonde falls with a parachute with a downward buoyancy and a downward velocity relative to the air parcel of about 5-6 m s$^{-1}$.

We have inserted this paragraph in the revised manuscript.

Line 144 - What error is associated with w and droplet collection efficiency (depending on size)?

$\rightarrow$ The ascending rate was typically ranging 4.0-6.0 m s$^{-1}$ during our launches performed at Concordia. So we can associate to $\omega$ an error (variability) of the order of ±1.0 m s$^{-1}$. This impacts on the SLWC calculation by ±3%.

The droplet collection efficiency $\varepsilon$ depends on the median droplet diameter considered. In Dexheimer et al. (2019), values of 11, 16 and 20 microns based on Lozowski et al. (1983) and Bain and Gayet (1982) were used to calculate SLWC. A median droplet diameter of 16 microns resulted in a collection efficiency greater than 0.9. This later value was finally given since it provided the lower estimate of SLWC in all observations performed in the Arctic. We thus also used an efficiency of 0.9 in our study.

The sensitivity of $\varepsilon$ to the median droplet diameter $d$ has thus been investigated. For $d$ varying from 11 to 20 microns, SLWC is varying by ±12%.

We have inserted these sensitivity studies in the revised version of the manuscript.

Bain, M. and Gayet, J.F.: Aircraft measurements of icing in supercooled and water droplet/ice crystal clouds. Journal of Applied Meteorology, 21, 631-641, https://www.jstor.org/stable/26180452, 1982.

Lozowski, E.P., Stallabrass, J.R. and Hearty, P.F.: The icing of an unheated, nonrotating cylinder. Part I: A simulation model. Journal of applied meteorology and climatology, 22(12), 2053-2062, https://doi.org/10.1175/1520-0450(1983)022%3C2053:TIOAUN%3E2.0.CO;2, 1983.

Line 152 - Why are the values integrated if the data is collected every 100 m?

→ It is not clear whether the reviewer refers to the vertical integral to get LWP or to the data analysis of the SLWC sonde. We clarified this point and modified the revised manuscript accordingly.

a) Data analysis.

The sampling rate of the SLWC is 0.2 s$^{-1}$, i.e. an observation is collected every 5 s, corresponding to ~25 m intervals along the vertical. We have applied a 4-point running average to all our observations. This means that our vertical profiles, even sampled every ~25 m, are not able to describe the variations for scales lower than 100 m.

b) LWP

As we explained above, calculating the LWP (expressed in g m$^{-2}$) from the SLWC (expressed in g m$^{-3}$) profile requires the integral along the vertical.

Line 180 - Is there a control for wet radon and fog events? How does the radiometer compare with LIDAR and radiosondes?

→ 1) We do not understand the point relative to "wet radon", maybe the reviewer is referring to "wet radome". Regarding the fog events, we have to remind that the radiometer is fully automated, thus there is no on line change in the operating mode depending on weather conditions. All the data processing is performed off line. The only external action on the radiometer (being given that it is installed inside a container) is the weekly cleanup of the protection window (maybe the radome to which we assume the reviewer is referring to) in case some snow deposition persists after heavy precipitation events that are very uncommon at Concordia.

2) It is the aim of the present study. We have clarified/synthetized this in the revised version, in the abstract and in the conclusions.

The three main outcomes from our analysis are:

a) In-situ observations of SLW clouds with SLWC sondes at Concordia station in Antarctica, first observations so far in Antarctica with a SLWC sonde. The location in height of the SLW clouds observed by the SLWC sonde is consistent with the profiles of humidity and temperature (and the deduced inflection points).

b) On average, the heights of the SLW clouds as observed by in-situ sondes and remonte-sensing LIDAR are consistent.

c) Liquid Water Path (vertically-integrated supercooled liquid water content) deduced by the sondes generally equals or is greater than LWP remotely sensed by a ground-based microwave radiometer.

Line 248 - "the profile of the SLWC" - SLWC is an integrated value - how do you define the profile?

→ The liquid content is usually referred to as Supercooled Liquid Water Content (SLWC) in unit of g m$^{-3}$. The Liquid Water Path (LWP) is the vertically-integrated SLWC in unit of g m$^{-2}$. Consequently, we can show the vertical profile of SLWC. If we integrate SLWC along the vertical, we obtain an LWP value that can be compared with the LWP observed by the HAMSTRAD radiometer. We again have clarified this point in the revised version of the manuscript.

Table I - Why does LIDAR sometimes have the SLW layer and other times not? Please add a new table for the clear air flight, it has different variables.

→ 1) See discussion presented above on the LIDAR sensitivity above an SLW cloud or liquid fog.

2) Done. We have created a new Table 2 that lists the information retrieved from the two launches performed in cloud-free conditions from which we can deduce bias and RMSE associated to SLWC and LWP, from the sonde and the radiometer, respectively. Note that Table 1 has been updated by including a new column highlighting the meteorological conditions ("Meteo") encountered for the launch in ascending and descending phases as: HP=Heavy Precipitation; LP=Light Precipitation; LF=Liquid Fog.

**Table 1:** List of SLW cloud flights performed during the 2021-2022 season over Concordia, together with date, launch time (UTC) and in italic the time (UTC) when the balloon hits the ground after the descent, SLW cloud vertical range (m) and associated LWP (g m$^{-2}$) in ascending (A) or descending (D) phase, considering only SLWC sonde observations above 400 m agl. Also shown are the SLW cloud vertical range (m) observed by the LIDAR in a ±1-hour time window centered on the start and, in italic, on the end of the flight. The LWP (g m$^{-2}$) shows the minimum-maximum values measured by HAMSTRAD for the same date over 24 hours. $H(\theta_{inf})$ is the height (m) of the inflection point in the vertical profile of potential temperature. Information is given on the type of string used (unwinder or unwound string of length $L$), on the velocity ω when it departs significantly from the nominal value of 5 m s$^{-1}$, and on surface liquid fog when present. Heights are always given in meters agl. Meteorological conditions (Meteo) synthetized as: HP=Heavy Precipitation; LP=Light Precipitation; LF=Liquid Fog.

| Launch A/D | Date YYMMDD | Launch Time HH:MM:SS UTC | Comments | Meteo | $H(\theta_{inf})$ m | SLW cloud vertical domain | | LWP g m$^{-2}$ | |
|---|---|---|---|---|---|---|---|---|---|
| | | | | | | Sonde m | LIDAR m | Sonde | Hamstrad Min-Max |
| L01 A | 211222 | 02:24:30 | Unwinder | HP | 710-750 | 400-500 | 400-600 700-750 | 7.37 | 2-10 |
| L03 D | 211225 | 08:53:15 *10:30:00* | Unwinder | HP | 950-1000 1450-1500 | 900-1000 1400-1500 | 600-800 *800-900* *1100-1200* | 3.67 | 2-6 |
| L04 A | 211225 | 15:48:51 | Unwinder | LP | 850-880 1400 1520 | 825-875 | 700-900 | 9.08 | 2-6 |
| L06 A | 211229 | 13:45:00 | $L = 40$ m $H > 750$ m | LP | $< 750$ | 750-825 | 500-800 | 7.48 | 1.0-3.5 |
| L07 A | 211229 | 17:47:51 | $L = 40$ m ω~3.5 m/s | LP | 700 850 | 425-600 750-900 | 600-750 | 33.17 23.94 | 1.0-3.5 |
| L14 A | 220124 | 13:51:05 | $L = 20$ m | LF | 630 900-920 1400 | 600 800-1000 | 50-250 750-850 | 575.35 | 1-5 |
| L14 | 220124 | 13:51:05 | $L = 20$ m | LF | 810 | | *50-300* | | 1-5 |

| | | | | | | | | |
|---|---|---|---|---|---|---|---|---|
| **D** | | *15:30:00* | | | 1340 1420 | 775-825(*) | 750-850 | 28.74 | |
| **L15 A** | 220128 | 06:08:27 | L = 20 m | LP | 650 910 1080 | 400-500 550-700 1000-1050 | 700-800 950-1050 | 17.62 13.75 7.31 | 2-5 |

(*) Most intense spike

**Table 2.** Flight L11 performed in cloud-free conditions during the 2021-2022 season over Concordia, together with date, launch time (UTC) and in italic the time (UTC) when the balloon hits the ground after the descent, in ascending (ASC) or descending (DES) phase. Also presented are: the LWP calculated from SLWC sonde observations, the minimum-maximum LWP (g m$^{-2}$) measured by HAMSTRAD for the same date over 24 hours, the variability $\sigma$ of the LWC as calculated from the SLWC sonde observations (g m$^{-3}$) and of the LWP as calculated from the HAMSTRAD observations (g m$^{-2}$). An information on the type of string used (unwinder or unwound string of length $L$) is also provided.

| Launch A/D | Date YYMMDD | Launch Time HH:MM:SS UTC | Comments | LWP g m$^{-2}$ | | Variability / $\sigma$ | |
|---|---|---|---|---|---|---|---|
| | | | | Sonde | Hamstrad Min-Max | SLWC Sonde g m$^{-3}$ | LWP Hamstrad g m$^{-2}$ |
| **L11 A** | 220117 | 06:35:15 | L = 40 m | ~0 | 0.4-1.0 | 0.08 | 0.2 |
| **L11 D** | 220117 | 06:35:15 *08:20:00* | L = 40 m | ~0 | 0.4-1.0 | 0.08 | 0.2 |

Line 304 - How do you calculate supersaturation with Vaisala radiosonde? What is the accuracy?

→ The supersaturation highlighted in the manuscript comes from the actual measurements. Relative Humidity (RH) provided by the Vaisala sonde is relative to liquid water. The RH values of 102 and 105% observed on 25 December 2022 (L03, Fig. 4) are provided by the Vaisala system. From The Vaisala White paper relative to the RS41 sondes (Vaisala Radiosonde RS41Measurement Performance, White Paper, Vaisala; available at: https://www.vaisala.com/sites/default/files/documents/WEA-MET-RS41-Performance-White-paper-B211356EN-B-LOW-v3.pdf), the accuracy of temperature and relative humidity are 0.3°C and 4%, respectively below 16 km altitude.

We have inserted this information in the revised manuscript.

Line 307 - Are the clouds 70 km from the radiometer similar to those observed on Concordia?

→ This is an interesting question. On 25 December 2021, we performed two launches at: a) 08:53:15 (L03) and b) 15:48:51 UTC (L04). For the first launch, we only processed the descending phase (L03 D) of the SLWC and PTU profiles at 10:30:00 UTC although we have available the PTU profiles performed in the ascending phase. Fig. R03 shows the profiles of wind direction and speed for the ascending and descending phases of L03, together with the presence of SLW clouds as observed by the LIDAR. At landing, 6000 s after launch, the balloon was 70 km to the North-East of Concordia (Fig. 3 top). Such a displacement is consistent with the wind direction and speed in the middle troposphere (250±20° and 18±4 m s$^{-1}$). In the lowermost troposphere the wind speed is much lower (< 5 m s$^{-1}$). Within the ±1-hour time window centered at the launch time, the LIDAR observed SLW clouds between 600 and 800

m (Fig. 2), that is to say just below the inflection point at 780 m corresponding to the 283-K isentrope. Then, 6000 s later and 70 km away from Concordia, in the descending phase, the SLWC sonde observed an SLW cloud between 900 and 1000 m encompassing the inflection point at 950 m corresponding also to the 283-K isentrope. Meanwhile, the LIDAR at Concordia observed two SLW clouds in the layers 800-900 and 1100-1200 m. Below 1600 m, the wind direction and speed were ranging 100-200° and 2-8 m s⁻¹, respectively.

So, 6000 s after the launch, considering the average wind speed (5 m s⁻¹) and orientation (100°, South-West) below 2000 m, an air parcel traveled only 30 km towards the West. As a consequence, for L03, the SLW clouds observed by the sonde 70 km away North-Eastward from the station in the descending phase has little chance to be the one observed by the LIDAR at the station in the ascending phase.

[Figure]

**Fig. R03:** Vertical profiles of the wind direction and speed measured by the Vaisala sonde in the ascending (left) and descending (right) phases of the launch L03 performed on 25 December 2021. The vertical extensions of the SLW clouds as observed by the LIDAR in a 2-hour time window centered on the launch time (ascending phase) or on the time of the flight end (descending phase) are highlighted in yellow or orange, respectively.

Later on, at 15:48:51 (L05), both the LIDAR and the SLWC sonde observed an SLW cloud in the range 700-900 m, encompassing or just below the inflection points at 850-880 m corresponding to the isentropes 281.5-282 K (Fig. 7). Therefore, it is very likely that the present SLW cloud is a remnant of (or the same as) the one observed 7 hours before over Concordia station within the 283-K isentrope.

We have inserted a new paragraph in the revised manuscript.

> An interesting point is to check whether the SLW cloud observed at 900-1000 m by the sonde 70 km away from the station in the descending phase (L03) is connected to the one observed 6000 s earlier by the LIDAR at the station at 600-800 m in the ascending phase, just below the inflection point at 780 m corresponding to the 283-K isentrope. In the ascending phase (Figure S26), the wind direction (250±20°) and the wind speed (18±4 m s⁻¹) in the middle troposphere are consistent with a balloon

travelling 70 km in the North-East direction in more than one hour and a half. On the other hand, in the lowermost troposphere (Figures S26 and S27), the wind is orientated to 120±20° and the wind speed is much lower (5±3 m s$^{-1}$). As a consequence, the probability for the SLW cloud observed by the SLWC sonde in the descending phase to be the one observed by the LIDAR in the ascending phase is very weak. Later on, at 15:48:51 (L05), both the LIDAR and the SLWC sonde in the ascending phase observed an SLW cloud in the range 700-900 m, encompassing or just below an inflection point at 850-880 m corresponding to the isentropes 281.5-282 K (Fig. 7). Therefore, it is very likely that the present SLW cloud is a remnant of (or the same as) the one observed 7 hours before over Concordia station within the 283-K isentrope.

Line 355 - The authors mention that L03 and L04 are consistent with LIDAR - where are the LIDAR data? Are they referring to Table I? The figure in the supplement (S2) shows only one layer.

→ See comments just above.

Figure 8—It was not cloudless because there was LWP all day. In fact, it was a day with thin clouds below the defined threshold.

→ The new Figure 8 shows the LIDAR observations (backscatter signal and depolarization ratio) for that day (17 January 2022) and do not exhibit any cloud or precipitation whatever its phase (liquid, solid or mixed).

[Figure]

**Figure 8.** Diurnal variation of the LIDAR backscatter signal (top, arbitrary unit) and depolarization ratio (middle, %) together with LWP from HAMSTRAD (bottom, black, scale on the left) and height of the SLW cloud (if any, red, scale on the right). Two vertical green dashed lines indicate 12:00 and 00:00 LT. The thick red vertical dashed lines indicate the time when balloon observations with SLWC sondes were performed in ascending (ASC) or descending (DES) phase.

In accordance with the LIDAR observations, the vertical profile of SLWC from the sonde in ascending and descending phases (Figs. 10 and 11, respectively) does not exhibit values significantly greater than the RMSE of 0.08 g m$^{-3}$ except at 425 m (0.11 g m$^{-3}$) in the ascending phase.

[Figure]

**Figure 10:** Same as Figure 6, but for 17 January 2022 at 06:35 UTC in ascending phase, in a cloud-free condition.

[Figure]

**Figure 11:** Same as Figure 6, but for 17 January 2022 at 06:35 UTC in descending phase, in a cloud-free condition.

Figure 10 - Many levels indicate liquid water (frequency peaks) during the flight - How was the LWP calculated on this day? How low relative humidity is associated with the line frequency?

→ 1) In cloud-free conditions, from df/dt values, we obtain the variability of the inferred SLWC values, hence the RMSE associated with the SLWC of the sonde.

2) We did not calculate LWP on that day since there were no SLW clouds.

3) The RH values confirm that no cloud layer was present on that period of time. So, the frequency profile recorded by the sonde translates as the noise of the signal.

Line 496 - The explanation for the possible error related to the fog that could stick to the wire is still not clear to me. What happens when the

→ The question seems to have been cut. Anyway, the point is that the presence of liquid fog close to the surface of the Concordia station has probably affected the vibration frequency of the sonde wire since liquid turned into ice at the surface of the wire prior to the launch. Consequently, we can naturally expect that the un-iced wire frequency (that is to say the frequency of the wire unaltered by ice) was not correct and affected the calculation of SLWC.

**Referee #2**

→ Specific changes have been made in response to the reviewer's comments and are described below. The reviewer's comments are recalled in blue.

The paper describes observations of supercooled liquid water clouds over Dome C, Antarctica with multiple instruments: balloon-borne Vaisala PTU and Anasphere SLW sondes, LIDAR, and HAMSTRAD microwave radiometer during field campaigns carried out in the period of 2021-2022. Generally supercooled liquid water clouds were observed near temperature inversion zone where atmospheric motion is capped. Both consistency and discrepancy among the different measurements were observed.

→ First of all, thank you for your positive feedback on the manuscript.

There are a few questions I would like the authors to address:

1. I totally get it that it is difficult to perform in-situ measurements over Antarctic and it is important that someone are doing it. On the other hand, as a scientific paper, can you emphasize the novelty and impact of this paper, i.e., what is the new findings of this paper that readers cannot get somewhere else?

→ The three main outcomes from our analysis are:

a) The in-situ observations of SLW clouds with SLWC sondes at Concordia station in Antarctica are the first observations so far in Antarctica with an SLWC sonde. The location in height of the SLW clouds observed by the SLWC sonde is consistent with the profiles of humidity and temperature (and the deduced inflection points).

b) On average, the heights of the SLW clouds as observed by in-situ sondes and remonte-sensing LIDAR are consistent.

c) The Liquid Water Path (LWP, the vertically-integrated supercooled liquid water content (SLWC)) deduced by the sondes generally equals or is greater than LWP remotely sensed by a ground-based microwave radiometer in spite of its low values (< 10 g m-2). Unfortunately, on some occasions far from nominal operation (surface liquid fog, low vertical ascent of the balloon), the sonde vertically-integrated SLWCs were overestimated by a factor of 5-10 compared to the HAMSTRAD LWPs.

We have highlighted these 3 points in the abstract and in the conclusions of the revised manuscript.

2. Under nominal conditions, the LWP values obtained by integrating SLWC sonde profiles are consistent with the HAMSTRAD measurements. Under non-ideal conditions, they are way off. The authors suggest the HAMSTRAD values are more trustworthy. Is this always the case? If yes, why don't we just perform HAMSTRAD observations all the time? What is the added value of doing SLWC sonde profiling?

→ The referee did not get the main point of the article, probably because we were not clear enough in the explanation. The radiometer HAMSTRAD is able to retrieve Liquid Water Path

(LWP) that is to say the vertically-integrated liquid water content (liquid water content is usually referred to as Supercooled Liquid Water Content = SLWC) at a 1-min time resolution. However, HAMSTRAD is not able to detect the height of the Supercooled Liquid Water (SLW) cloud. Conversely, the LIDAR is able to retrieve the vertical distribution of SLW cloud but not the SLWC values within the cloud. The sonde we have used combine the two main advantages of the radiometer and of the LIDAR, that is to say the detection of the vertical distribution of SLW clouds and SLWC within the cloud.

We have synthetized the pros and cons of the three instruments in Table 3. (see replies to the comments of the Reviewer#1).

3. Figure 1 is not informative at all, which could be replaced by a cartoon drawing showing how things work.

→ We partly agree with this point. Figure 1 is, in our opinion, very informative since it clearly shows the great difficulty of handling 2 different sondes (Vaisala PTU and SLWC sondes) simultaneously in order to minimize the oscillations inferred by the launch itself and by the handling of the sondes. We have underlined this point more clearly in the revised version of the manuscript by detailing the methodology employed to launch the 2 sondes simultaneously and modified the incriminated Figure.

[Figure]

**Figure 1.** The methodology employed to launch both the SLWC and the PTU sondes with meteorological balloons is synthetized as follow. 1) The Vaisala PTU sondes are calibrated into the quiet building of the Concordia station at room temperature using the standard Digicora ground-check system. 2) The SLWC sonde is connected to the PTU sonde at room temperature and then is transported outdoors to the meteorological shelter. The two sondes are attached to the meteorological balloon after inflation of the balloon. 3) Then, after leaving the shelter, a person holds the SLWC sonde in his/her hands while another holds both the meteorological balloon and the PTU sonde. When the meteorological and technical conditions are optimised, the balloon is launched. The picture represents a launch of a Vaisala PTU sonde (left hand of the man in blue) and an Anasphere SLWC sonde (right hand of the man in red) attached to the Totex TA100 meteorological balloon, together with the red parachute and the unwinder for the first flight on 22 December 2021.

4. What is the functionality of unwound string or unwinder? It is not clear.

→ For optimal operation, the PTU sondes should be a few tens of meters away from the balloon. The use of an unwinder copes with inconvenience of having a long string to manage before launching and reduce the pulling forces at the start of ascent until the balloon has reached its nominal ascent rate. Nevertheless, unwinding after launch produces a pendular rotation of the sonde until the unwinding terminates. But only the SLWC sonde is affected by the rotation. Concretely, we have noticed that an unwinder stabilizes the SLWC sondes more rapidly than no unwinder. On average, it takes about one minute for the SLWC sonde to stabilize, thus with an ascending velocity of 5 m s$^{-1}$, the first 300 m height cannot be scientifically exploitable. We have enlarged the vertical threshold to 400 m height in our study in order to minimize as much as possible the impact of the launch to the SLWC sonde stabilization.

We have clarified this point in the revised version.